# DirectEdit: Step-Level Accurate Inversion for Flow-Based Image Editing

**Desong Yang** [1]    **Mang Ye** [1]

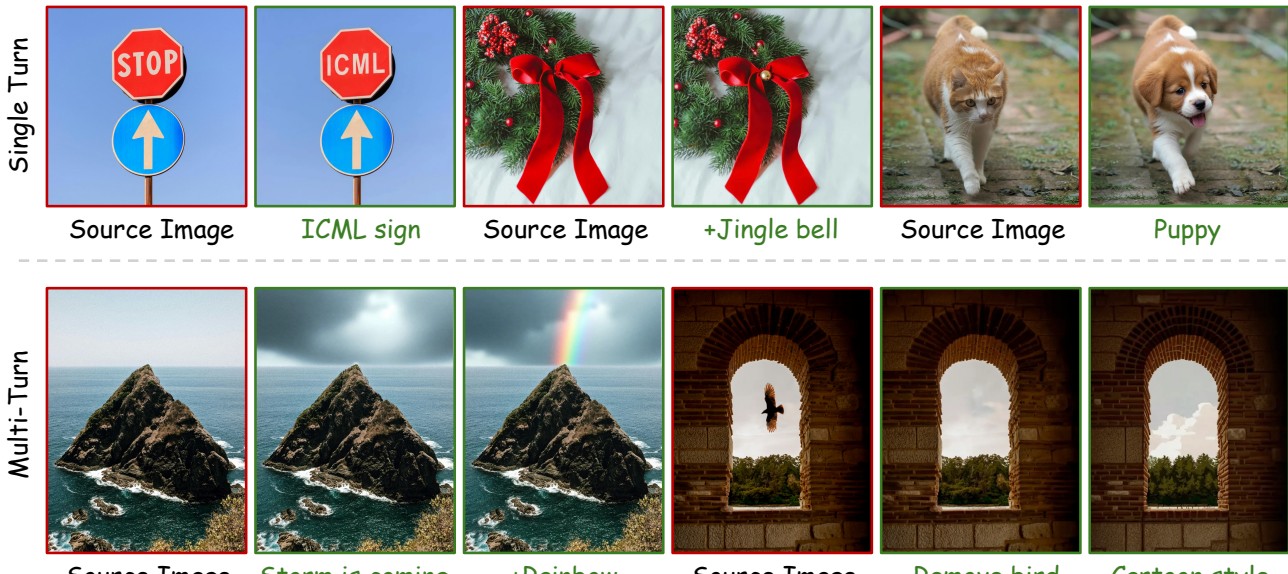

*Figure 1.* We present **DirectEdit**, a simple yet highly effective training-free method for flow-based image editing. Compared with existing inversion-based approaches, DirectEdit explicitly aligns the reconstruction and inversion trajectories and introduces an effective latent feature interaction mechanism, enabling step-level accurate reconstruction and precise background preservation. Extensive experiments across diverse editing tasks demonstrate that DirectEdit achieves state-of-the-art performance.

## Abstract

With recent advancements in large-scale pre-trained text-to-image (T2I) models, training-free image editing methods have demonstrated remarkable success. Typically, these methods involve adding noise to a clean image via an inversion process, followed by separate denoising steps for the reconstruction and editing paths during the forward process. However, since the reconstruction path is approximated using noisy latents from mismatched timesteps, existing methods inevitably suffer from accumulated drift, which fundamentally limits reconstruction fidelity. To address this challenge, we systematically analyze the inversion process within the flow transformer and

propose DirectEdit, a simple yet effective editing method that eliminates the inherent reconstruction error without introducing additional neural function evaluations (NFEs). Unlike most prior works that attempt to rectify the inversion path, DirectEdit focuses on directly aligning the forward paths, enabling precise reconstruction and reliable feature sharing. Furthermore, we introduce a preservation mechanism based on attention feature injection and multi-branch mask-guided noise blending, which effectively balances fidelity and editability. Extensive experiments across diverse scenarios demonstrate that DirectEdit achieves efficient and accurate image editing, delivering superior performance that outperforms state-of-the-art methods. Code and examples are available at https://desongyang.github.io/Directedit.

[1]National Engineering Research Center for Multimedia Software, School of Computer Science, Wuhan University, Wuhan, China. Correspondence to: Mang Ye <yemang@whu.edu.cn>.

*Proceedings of the 43rd International Conference on Machine Learning*, Seoul, South Korea. PMLR 306, 2026. Copyright 2026 by the author(s).

## 1. Introduction

In recent years, models based on Rectified Flow (RF) (Lipman et al., 2022; Liu et al., 2022) have achieved significant

success in large-scale text-to-image generation, enabling diverse image editing capabilities through flexible text guidance. Instruction-based methods (Liu et al., 2025; Wu et al., 2025; Cai et al., 2025) typically finetune pre-trained models by collecting paired datasets of original and edited images along with editing instructions. However, due to the inherent difficulty in constructing large-scale, high-quality datasets, these approaches are computationally expensive and prone to introducing biases. In contrast, training-free methods (Rout et al., 2024; Wang et al., 2024; Kulikov et al., 2025; Deng et al., 2024; Xu et al., 2025; Xie et al., 2025; Zhu et al., 2025), which solely leverage the priors of pre-trained generative models to perform editing, have garnered widespread research attention. Nevertheless, the majority of these methods rely on an inversion process, where the noisy latent at the current timestep is used to approximate the latent of the subsequent timestep. This approximated latent is then employed to estimate the reversed velocity field, resulting in a deviation from the forward process and ultimately leading to an inevitable accumulation of reconstruction errors.

Existing RF-based training-free methods primarily focus on mitigating approximation errors within the inversion path. RF-inversion (Rout et al., 2024) constructs a conditional vector field for inversion and editing based on dynamic optimal control derived from the Linear Quadratic Regulator (LQR). Subsequent methods (Wang et al., 2024; Deng et al., 2024) utilize high-order Ordinary Differential Equation (ODE) solvers to further reduce approximation errors. FTEdit (Xu et al., 2025) minimizes errors via fixed-point iteration optimization during the inversion process and corrects the reconstruction trajectory after each denoising step. DNAEdit (Xie et al., 2025) estimates the velocity of the next timestep through interpolation and continuously optimizes the Gaussian noise used for this interpolation. Recently, inversion-free methods such as FlowEdit (Kulikov et al., 2025; Kim et al., 2025; Yang et al., 2025b) have been proposed. These methods attempt to mitigate approximation errors by constructing a direct path from the source image to the edited image via interpolation with random noise. Although offering marginal improvements over standard Euler methods, these training-free methods either continue to suffer from error accumulation or struggle to guarantee the fidelity of the generated images.

In this work, we investigate the aforementioned issues by conducting a deep exploration of the inversion and editing processes within the Rectified Flow (RF) framework. Fundamentally, rather than achieving mere pixel-level reconstruction, our primary concern lies in the correctness of the source features injected into the edited image. We observe that existing editing methods fail to fundamentally address the issue of error accumulation. Although these approaches significantly mitigate the deviation of the reconstruction trajectory from the inversion path, the persistence

of errors at each denoising step means the editing path is continuously injected with "drifted" source features. This inevitably degrades source feature preservation and leads to visible artifacts, as illustrated in Figure 2.

This raises a critical question: How can we achieve step-level accurate reconstruction? To answer this question, we propose DirectEdit, a simple yet highly effective novel method. Distinguished from most approaches that attempt to rectify the inversion path, DirectEdit aligns the forward process directly with the inversion path step-by-step. This enables precise reconstruction and reliable feature sharing without introducing additional Neural Function Evaluations (NFEs). To better balance editability and fidelity, we further inject value features from the source branch during selected denoising steps, thereby preserving the semantic information of the original object. Additionally, we propose a multi-branch mask-guided noise blending mechanism leveraging Multimodal Large Language Models (MLLMs) (Yang et al., 2025a). This mechanism generates mask-blended noisy latents tailored to specific editing types, facilitating precise background preservation.

In summary, our contributions are as follows:

- We propose DirectEdit, a simple yet effective training-free image editing framework. To the best of our knowledge, it is the first inversion-based editing method that achieves step-level accurate reconstruction.

- We introduce a novel preservation mechanism combining multi-branch mask-guided noise blending with attention-based feature injection to effectively balance fidelity and editability.

- We conduct extensive experiments on various flow models, demonstrating that DirectEdit achieves state-of-the-art performance across diverse editing tasks.

## 2. Related Work

### 2.1. Image Editing

Existing image editing methodologies can be broadly categorized into two classes: training-based (Brooks et al., 2023; Zhang et al., 2023; Wu et al., 2025; Cai et al., 2025) and training-free (Wang et al., 2024; Kulikov et al., 2025; Deng et al., 2024; Xu et al., 2025; Xie et al., 2025; Zhu et al., 2025) approaches. Training-based methods typically train end-to-end image editing models by collecting large-scale datasets. However, owing to the scarcity of high-quality editing data, these approaches are computationally expensive and susceptible to introducing biases. In contrast, training-free methods have garnered increasing attention due to their efficiency and practicality. These methods generally commence

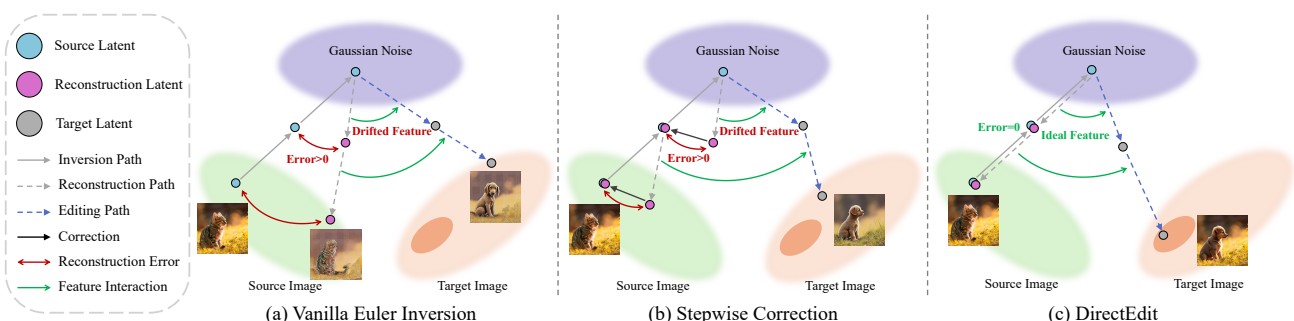

*Figure 2.* **Comparison of Inversion Methods. (a)** Standard Euler inversion. Due to the accumulation of approximation errors between the reconstruction and inversion paths, it fails to accomplish successful reconstruction and editing. **(b)** Inversion via stepwise correction. Although error accumulation is mitigated, the persistence of step-level reconstruction errors results in the continuous injection of "drifted" features, leading to unsatisfactory editing outcomes. **(c)** Our proposed DirectEdit. By ensuring strict alignment between the reconstruction path and the inversion trajectory, DirectEdit facilitates ideal feature interaction, thereby achieving superior editing performance.

with an inversion process that maps a real image to its corresponding noisy latent, followed by a forward process that transforms this latent into an edited image. Early diffusion-based approaches (Hertz et al., 2022; Cao et al., 2023; Wang & Ye, 2024; Dong & Ye, 2025) typically employed DDIM inversion (Song et al., 2020), predicated on the assumption that the Ordinary Differential Equation (ODE) process can be reversed in the limit of small steps. Although numerous studies have attempted to mitigate approximation errors in DDIM inversion, step-level errors persist, ultimately resulting in the drift of injected features and error accumulation. Rectified Flow (RF) (Liu et al., 2022) has recently achieved remarkable success in the generative domain; however, its inversion remains challenging and underexplored.

### 2.2. RF-based Training-free Editing

To mitigate the approximation errors inherent in the standard Euler inversion used in RF models, various strategies have been proposed to refine the inversion path. RF-Inversion (Rout et al., 2024) introduces an additional vector field for inversion and editing based on dynamic optimal control. Subsequent approaches (Wang et al., 2024; Deng et al., 2024) leverage high-order ODE solvers to reduce numerical errors. FTEdit (Xu et al., 2025) employs fixed-point iteration to optimize noisy latents during inversion, and corrects the reconstruction trajectory after each denoising step. More recently, DNAEdit (Xie et al., 2025) estimates the velocity of the next timestep via interpolation and continuously optimizes the random noise used for this interpolation. Nevertheless, both prior research (Xu et al., 2025) and our empirical findings indicate that Euler inversion for flow-based transformers is significantly more sensitive to approximation errors compared to DDIM inversion, thereby exacerbating the phenomenon of error accumulation.

Recently, inversion-free methods such as FlowEdit (Kulikov et al., 2025; Kim et al., 2025; Yang et al., 2025b) have emerged. These approaches construct a direct trajectory

from the source to the edited image by interpolating between the image and random noise at each step to compute velocity differences, thereby circumventing approximation errors associated with inversion. However, due to the stochastic nature of the interpolated noise, these methods introduce substantial errors, making it difficult to achieve satisfactory fidelity. Furthermore, by initiating the process directly from the image rather than an inverted latent, they inadvertently compromise editing flexibility and practicality. In contrast, our work aims to explicitly eliminate the approximation errors introduced during the inversion process.

## 3. Method

### 3.1. Preliminaries

**Rectified Flow.** The Rectified Flow (Liu et al., 2022) model linearly interpolates the probability path between two observed distributions $\mathbf{Z}_0 \sim \pi_0$ and $\mathbf{Z}_1 \sim \pi_1$:

$$\mathbf{Z}_t = t\mathbf{Z}_1 + (1-t)\mathbf{Z}_0, \quad t \in [0,1], \tag{1}$$

where $\pi_0$ is chosen as the standard Gaussian distribution $\mathcal{N}(\mathbf{0}, \mathbf{I})$, and $\pi_1$ denotes the image distribution. It learns this straight probability transport path via the ordinary differential equation (ODE) in Equation (2):

$$\mathrm{d}\mathbf{Z}_t = v_\theta(\mathbf{Z}_t)\mathrm{d}t, \quad t \in [0,1], \tag{2}$$

where $v_\theta(\cdot)$ represents the velocity field parameterized by neural network weights $\theta$. The training objective is to regress the velocity field via the least-squares loss:

$$\mathcal{L} = \mathbb{E}_{t\sim\mathcal{U}[0,1],\mathbf{Z}_1\sim\pi_1} \left[ \|(\mathbf{Z}_1 - \mathbf{Z}_0) - v_\theta(\mathbf{Z}_t)\|^2 \right]. \tag{3}$$

The sampling process initiates from $\mathbf{Z}_0$ and integrates the ODE from $t = 0 \to 1$ via the Euler method to generate the data sample $\mathbf{Z}_1$, as formulated in Equation (4):

$$\mathbf{Z}_{t+1} = \mathbf{Z}_t + (\sigma_{t+1} - \sigma_t)v_\theta(\mathbf{Z}_t), \tag{4}$$

where the time interval $[0, 1]$ is discretized into $T$ steps, denoted as $\{\sigma_0, \ldots, \sigma_T\}$.

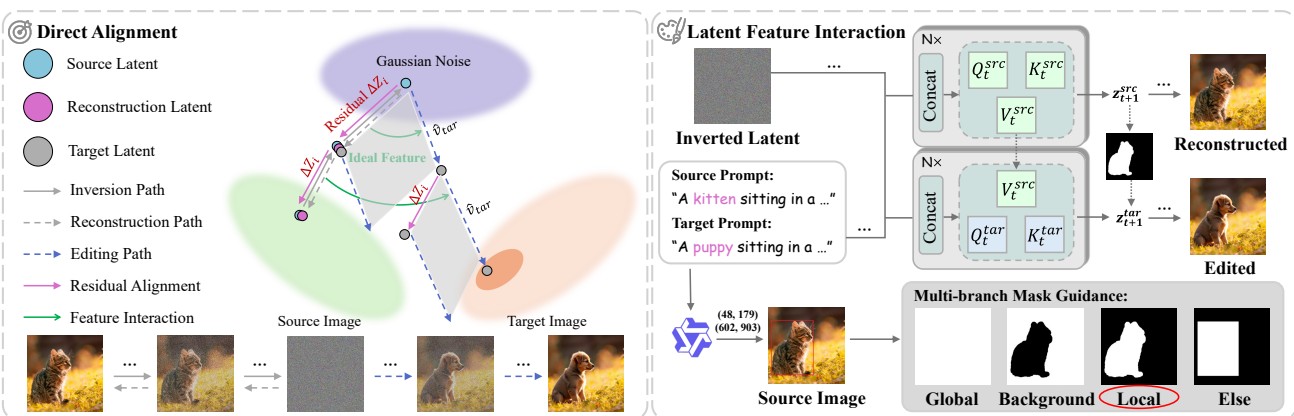

*Figure 3.* **Overview of DirectEdit. Left: Direct Alignment for Accurate Inversion.** By explicitly aligning with the inversion trajectory, we achieve step-level accurate reconstruction, thereby facilitating the extraction of ideal source image features. **Right: Latent Feature Interaction.** We further introduce a preservation mechanism that leverages noisy latents and attention features from the reconstruction path. This mechanism enables flexible and diverse editing while ensuring precise background preservation.

**Approximation Errors in Inversion.** Unlike the forward process which maps the standard Gaussian distribution $\pi_0$ to the image distribution $\pi_1$, the inversion process maps $\pi_1$ back to $\pi_0$. Based on Equation (4), the ideal form of Euler inversion can be easily derived as follows:

$$\mathbf{Z}_t^{inv} = \mathbf{Z}_{t+1}^{inv} - (\sigma_{t+1} - \sigma_t)v_\theta(\mathbf{Z}_t^{inv}). \quad (5)$$

However, since the current state is $\mathbf{Z}_{t+1}^{inv}$, the value of $v_\theta(\mathbf{Z}_t^{inv})$ is inaccessible. Similar to DDIM inversion (Song et al., 2020), most existing RF inversion methods approximate $v_\theta(\mathbf{Z}_t^{inv})$ using $v_\theta(\mathbf{Z}_{t+1}^{inv})$, based on the premise that the difference between noisy latents at adjacent timesteps is relatively small:

$$\mathbf{Z}_t^{inv} = \mathbf{Z}_{t+1}^{inv} - (\sigma_{t+1} - \sigma_t)v_\theta(\mathbf{Z}_{t+1}^{inv}). \quad (6)$$

Although the single-step error of this approach is relatively minor, the approximation error accumulates gradually as the number of denoising steps increases, eventually leading to a complete deviation from the inversion path.

A prevalent strategy is to perform stepwise correction during the forward process (Ju et al., 2023; Xu et al., 2025). Specifically, by enforcing $\mathbf{Z}_t = \mathbf{Z}_t^{inv}$ in the reconstruction path after each denoising step, the accumulation of reconstruction errors can be mitigated. However, we observe that this approach fails to fundamentally address the issue. Due to the persistence of step-level reconstruction errors, the source image features injected during each denoising step are inherently "drifted". While this method performed well in the context of DDIM inversion, the sensitivity of the Euler inversion used in RF to approximation errors exacerbates the impact of single-step errors, as illustrated in Figure 2.

### 3.2. Direct Alignment for Accurate Inversion

Rather than seeking an inversion trajectory that perfectly aligns with the reconstruction path, DirectEdit dedicates itself to aligning the reconstruction path with the inversion trajectory (see Figure 3). Specifically, regarding the inversion trajectory as a pivot, our objective is to ensure that the reconstruction path perfectly aligns with the inversion trajectory, thereby eliminating step-level reconstruction errors. Assuming the current denoising timestep is $t$, this condition necessitates the satisfaction of the following equation:

$$\mathbf{Z}_{t+1} = \mathbf{Z}_{t+1}^{inv}. \quad (7)$$

Given that the adjacent discrete time difference $\Delta\sigma = \sigma_{t+1} - \sigma_t$ is identical for both the inversion and forward processes, and $\mathbf{Z}_0 = \mathbf{Z}_0^{inv}$, based on Equation (4) and Equation (6), Equation (7) can be reformulated as:

$$v_\theta(\mathbf{Z}_t) = v_\theta(\mathbf{Z}_{t+1}^{inv}). \quad (8)$$

Note that $\mathbf{Z}_{t+1}^{inv}$ is accessible during the inversion process. We directly map the source image $\mathbf{Z}_1$ to the noisy latent $\mathbf{Z}_0$ via the standard Euler inversion process shown in Equation (6). Crucially, we record the latent residual $\Delta\mathbf{Z}_t$ at each noise addition step. During the forward process, we first add this latent residual to $\mathbf{Z}_t$ to obtain $\hat{\mathbf{Z}}_t$, thereby aligning it with the inversion path velocity:

$$\Delta\mathbf{Z}_t = \mathbf{Z}_{t+1}^{inv} - \mathbf{Z}_t^{inv}, \quad \hat{\mathbf{Z}}_t = \mathbf{Z}_t + \Delta\mathbf{Z}_t. \quad (9)$$

Subsequently, predicting the velocity using $\hat{\mathbf{Z}}_t$ ensures complete alignment with the velocity in the inversion process. We then utilize this velocity to update $\mathbf{Z}_t$:

$$\mathbf{Z}_{t+1} = \mathbf{Z}_t + (\sigma_{t+1} - \sigma_t)v_\theta(\hat{\mathbf{Z}}_t). \quad (10)$$

This achieves precise reconstruction from $\mathbf{Z}_t$ to $\mathbf{Z}_{t+1}$, facilitating more accurate feature injection. The algorithm is presented in Algorithm 1. For further details, we provide a detailed theoretical analysis and comparisons with existing inversion methods in Appendix A.

## 3.3. Latent Feature Interaction

**Multi-branch Mask-guided Noise Blending.** In real-world image editing tasks, maintaining the semantic integrity and structural layout of the original image is paramount. While DirectEdit achieves precise reconstruction by aligning inversion trajectories, the editing process remains susceptible to unintended modifications induced by the target prompt. To mitigate this, we propose a *Multi-branch Mask-guided Noise Blending* mechanism. Specifically, we feed the source image and textual prompts into a Multimodal Large Language Model (MLLM) to determine the editing type $O$ and query the coordinate pairs defining the region of interest, denoted as $P(x_1, y_1)$ and $Q(x_2, y_2)$. Subsequently, utilizing the Segment Anything Model (SAM) (Kirillov et al., 2023), we generate the editing region mask $\mathcal{M}$ based on the identified editing type:

$$\mathcal{M}(O, P, Q) = \begin{cases} \mathcal{S}(\mathcal{B}(P, Q)) & \text{if } O = \text{Local} \\ 1 - \mathcal{S}(\mathcal{B}(P, Q)) & \text{if } O = \text{Background} \\ 1 & \text{if } O = \text{Global} \\ \mathcal{B}(P, Q) & \text{otherwise,} \end{cases} \tag{11}$$

where $\mathcal{S}(\cdot)$ denotes the binary segmentation operation performed by SAM, and $\mathcal{B}(P, Q)$ represents the rectangular bounding box parallel to the axes defined by the diagonal points $P$ and $Q$. This multi-branch mask generation process ensures tailored editing for diverse editing types. Specifically, for **local editing**, we perform segmentation to obtain the binary mask of the target object. In the case of **background editing**, we derive the mask by taking the complement of the object mask. For **global editing** scenarios (e.g., style transfer), we regard the entire image as the editable region. Finally, regarding **other editing types** such as object addition, we directly utilize the rectangular bounding box as the region of interest.

Leveraging this automatically generated mask $\mathcal{M}$, we blend the noisy latents from the reconstruction path and the editing path to achieve precise background preservation:

$$\mathbf{Z}_t^{tar} = \mathbf{Z}_t^{src} \odot (1 - \mathcal{M}) + \mathbf{Z}_t^{tar} \odot \mathcal{M}, \tag{12}$$

where $\odot$ denotes element-wise multiplication. Notably, the mask $\mathcal{M}$ can also be manually provided by the user, ensuring editing flexibility and user-friendliness. Capitalizing on the robust reconstruction capabilities of DirectEdit, this blending process guarantees high fidelity in non-edited regions, thereby effectively preventing unintended modifications induced by the target prompt.

**Attention Feature Injection.** To further preserve fine-grained details of the edited object, drawing inspiration from prior studies (Hertz et al., 2022; Xu et al., 2025), we manipulate the self-attention layer by injecting the *Value*

---

**Algorithm 1** DirectEdit Algorithm

**Input:** Source image $\mathbf{I}_{src}$, Source text $\psi_{src}$, Target text $\psi_{tar}$, Timesteps $\{\sigma_0, \ldots, \sigma_T\}$, Denoising steps $T$.
$\mathbf{Z}_T \leftarrow \text{VAE\_Encode}(\mathbf{I}_{src})$      ▷ *Inversion Process*
**for** $t = T - 1$ **to** $0$ **do**
    $\mathbf{Z}_t^{inv} \leftarrow \mathbf{Z}_{t+1}^{inv} - (\sigma_{t+1} - \sigma_t) v_\theta(\mathbf{Z}_{t+1}^{inv}, \psi_{src})$
    $\Delta \mathbf{Z}_t \leftarrow \mathbf{Z}_{t+1}^{inv} - \mathbf{Z}_t^{inv}$
**end for**

$\mathbf{Z}_0^{src}, \mathbf{Z}_0^{tar} \leftarrow \mathbf{Z}_0^{inv}$      ▷ *Editing Process*
$O, P, Q \leftarrow \text{MLLM}(\mathbf{I}_{src}, \psi_{src}, \psi_{tar})$
$\mathcal{M} \leftarrow \mathcal{M}(O, P, Q)$
**for** $t = 0$ **to** $T - 1$ **do**
    $\hat{\mathbf{Z}}_t^{src} \leftarrow \mathbf{Z}_t^{src} + \Delta \mathbf{Z}_t, \quad \hat{\mathbf{Z}}_t^{tar} \leftarrow \mathbf{Z}_t^{tar} + \Delta \mathbf{Z}_t$
    $\mathbf{Z}_{t+1}^{src} \leftarrow \mathbf{Z}_t^{src} + (\sigma_{t+1} - \sigma_t) v_\theta(\hat{\mathbf{Z}}_t^{src}, \psi_{src})$
    $\hat{v}_t^{tar} \leftarrow v_\theta(\hat{\mathbf{Z}}_t^{tar}, \psi_{tar})\{\mathbf{F}_t^{tar} \leftarrow \hat{\mathbf{F}}_t^{tar}\}$
    $\mathbf{Z}_{t+1}^{tar} \leftarrow \mathbf{Z}_t^{tar} + (\sigma_{t+1} - \sigma_t) \hat{v}_t^{tar}$
    $\mathbf{Z}_{t+1}^{tar} \leftarrow \mathbf{Z}_{t+1}^{src} \odot (1 - \mathcal{M}) + \mathbf{Z}_{t+1}^{tar} \odot \mathcal{M}$
**end for**
**Output:** $\text{VAE\_Decode}(\mathbf{Z}_T^{src}), \text{VAE\_Decode}(\mathbf{Z}_T^{tar})$

---

features from the reconstruction path into the editing path:

$$\hat{\mathbf{F}}_t^{tar} = \begin{cases} \text{Attention}(\mathbf{Q}_t^{tar}, \mathbf{K}_t^{tar}, \mathbf{V}_t^{src}) & \text{if } t < t_{inj} \\ \text{Attention}(\mathbf{Q}_t^{tar}, \mathbf{K}_t^{tar}, \mathbf{V}_t^{tar}) & \text{otherwise,} \end{cases} \tag{13}$$

where $\hat{\mathbf{F}}_t^{tar}$ denotes the output features of the self-attention module within the MM-DiT at timestep $t$, and $t_{inj}$ serves as a hyperparameter governing the duration of the injection. By substituting the original features $\mathbf{F}_t^{tar}$ of the editing path with the derived attention features $\hat{\mathbf{F}}_t^{tar}$, we obtain the refined target velocity $\hat{v}_t^{tar}$:

$$\hat{v}_t^{tar} = v_\theta(\hat{\mathbf{Z}}_t^{tar}, \psi_{tar})\{\mathbf{F}_t^{tar} \leftarrow \hat{\mathbf{F}}_t^{tar}\}, \tag{14}$$

where $\hat{\mathbf{Z}}_t^{tar}$ denotes the target latent estimated via Direct Alignment, and $\psi_{tar}$ represents the editing prompt. The primary objective of this step focuses on the refinement within the mask interior (i.e., the editable region), aiming to preserve the semantic details of the original object. Subsequently, we utilize this refined velocity $\hat{v}_t^{tar}$ to update $\mathbf{Z}_t^{tar}$. Regarding specific architectures, for SD3.5 (Esser et al., 2024), we apply injection across all MM-DiT blocks except the final one. Conversely, for FLUX (Labs, 2024), we empirically find that injecting solely into the single blocks suffices, as these modules simultaneously extract information from both the source image and text prompts, thereby enhancing the model's capacity to retain source features. Benefiting from the precise reconstruction afforded by DirectEdit, the injected unbiased features can more accurately reflect the authentic information of the original image.

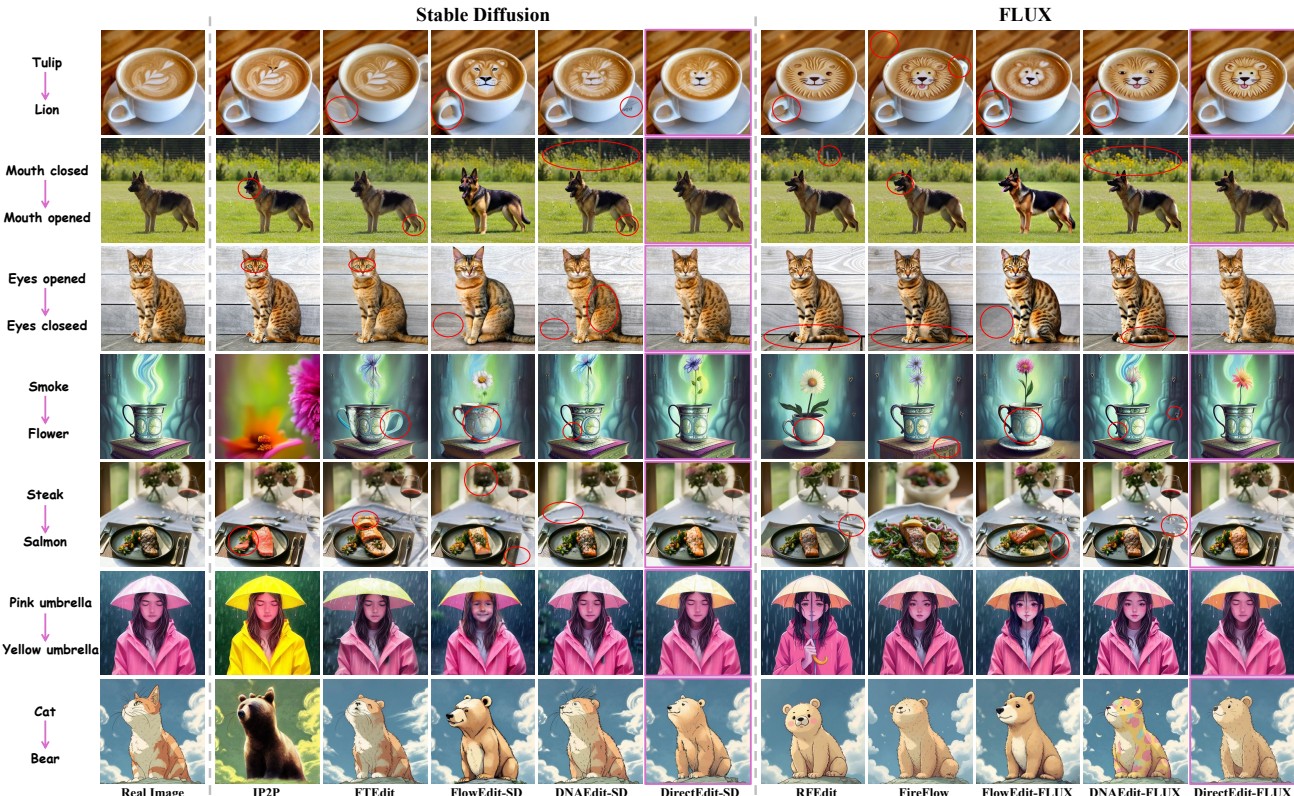

*Figure 4.* **Qualitative comparison** with various editing methods. Our method demonstrates exceptional performance across a diverse range of editing tasks, outperforming prior state-of-the-art approaches in terms of both background preservation and text alignment.

# 4. Experiments

## 4.1. Setup

**Evaluation Datasets and Metrics.** We evaluate our proposed method and all baselines on the PIE-Bench dataset (Ju et al., 2023) for image editing tasks. This benchmark comprises 700 images covering 9 representative editing categories, evenly distributed across natural and synthetic scenes. Our evaluation metrics concurrently focus on three pivotal aspects: structure retention, background preservation and text-image consistency. Specifically, we utilize Structure Distance (Ju et al., 2023) to assess structure retention. To evaluate the background fidelity of the generated images, we employ Mean Squared Error (MSE), Peak Signal-to-Noise Ratio (PSNR), Learned Perceptual Image Patch Similarity (LPIPS) (Zhang et al., 2018), and Structural Similarity Index (SSIM) (Wang et al., 2004). Furthermore, we adopt CLIP Similarity (Wu et al., 2021) to quantify the semantic consistency between the target prompt and the edited image.

**Implementation Details.** We implement our method utilizing FLUX.1-dev (Labs, 2024) and SD3.5-medium (Esser et al., 2024) as the backbone models, respectively. For both variants, the number of denoising steps is set to 30. We uniformly configure the Classifier-Free Guidance (CFG) (Ho

& Salimans, 2022) scale to 1 for the inversion process and 2 for the editing process. We primarily benchmark DirectEdit against prior state-of-the-art training-free image editing approaches, encompassing both representative diffusion-based methods (Brooks et al., 2023; Hertz et al., 2022) and the latest RF-based techniques (Rout et al., 2024; Deng et al., 2024; Kulikov et al., 2025; Xu et al., 2025; Xie et al., 2025). All baselines are evaluated using their officially recommended hyperparameters. More detailed information for experiment setup is provided in Appendix D.

## 4.2. Main Results

**Quantitative Comparison on Real Image Editing.** We conduct a quantitative comparison of our method against several state-of-the-art image editing approaches (Brooks et al., 2023; Hertz et al., 2022; Rout et al., 2024; Deng et al., 2024; Xu et al., 2025; Kulikov et al., 2025; Xie et al., 2025; Kim et al., 2025) on the PIE-Bench (Ju et al., 2023). We evaluate the editing performance across three dimensions: structure retention, background preservation, and CLIP Similarity. Additionally, the final column of Table 1 presents a comprehensive ranking for each method, derived by averaging the rankings across these metrics. As shown in Table 1, our method excels in balancing editability with the preser-

*Table 1.* **Quantitative comparisons** on PIE-Bench. The top three results are highlighted in red , blue , and green , respectively.

| Baselines | | Structure | Background Preservation | | | | CLIP Similarity | | Rank |
|---|---|---|---|---|---|---|---|---|---|
| Method | Model | Distance ↓ | PSNR ↑ | LPIPS ↓ | MSE ↓ | SSIM↑ | Whole ↑ | Edited ↑ | Avg. ↓ |
| IP2P | SD1.5 | 58.13 | 20.95 | 159.20 | 230.87 | 76.39 | 23.61 | 21.77 | 14.43 |
| P2P | SD1.5 | 15.44 | 27.52 | 59.69 | 34.20 | 84.41 | 24.75 | 21.01 | 7.86 |
| RF-Inversion | FLUX | 41.17 | 20.86 | 187.01 | 120.12 | 71.21 | 25.08 | 22.39 | 13.57 |
| RFEdit | FLUX | 25.15 | 24.33 | 121.59 | 56.98 | 82.84 | 25.57 | 22.54 | 9.14 |
| FireFlow | FLUX | 27.40 | 23.11 | 128.46 | 70.75 | 81.30 | 26.13 | 22.87 | 9.43 |
| FlowEdit | FLUX | 27.83 | 21.96 | 112.15 | 94.94 | 83.40 | 25.26 | 22.60 | 10.57 |
| DNAEdit | FLUX | 16.81 | 25.20 | 86.68 | 48.35 | 87.21 | 24.81 | 22.12 | 7.71 |
| Ours (w/o mask) | FLUX | 21.93 | 24.70 | 102.92 | 56.76 | 85.76 | 25.89 | 22.74 | 6.86 |
| Ours | FLUX | 17.94 | 32.63 | 35.45 | 25.05 | 93.49 | 25.39 | 22.45 | 4.00 |
| FTEdit | SD3.5 | 21.06 | 23.49 | 90.25 | 61.78 | 86.23 | 25.21 | 21.78 | 9.29 |
| FlowEdit | SD3.5 | 23.13 | 23.29 | 92.81 | 69.09 | 85.22 | 26.71 | 23.59 | 7.29 |
| FlowAlign | SD3.5 | 33.49 | 24.06 | 68.91 | 54.29 | 86.33 | 25.64 | 22.40 | 8.00 |
| DNAEdit | SD3.5 | 11.03 | 27.71 | 60.51 | 26.28 | 90.13 | 25.20 | 22.25 | 5.14 |
| Ours (w/o mask) | SD3.5 | 15.23 | 26.18 | 67.28 | 34.00 | 88.75 | 26.13 | 22.50 | 4.29 |
| Ours | SD3.5 | 14.65 | 31.82 | 31.36 | 21.64 | 92.28 | 25.64 | 22.67 | 2.43 |

vation of non-edited regions. In comparison, DNAEdit (Xie et al., 2025) and P2P (Hertz et al., 2022) exhibits strong structural preservation but suffers from under-editing. Conversely, while FlowEdit (Kulikov et al., 2025) and FireFlow (Deng et al., 2024) yield higher CLIP scores, this advantage comes at the expense of compromised background and structural fidelity. Our method effectively bridges this gap, demonstrating exceptional background preservation without constraining editability, thereby highlighting its capacity for fine-grained editing tasks. Furthermore, we also conducted a quantitative comparison under various hyperparameter settings. As illustrated in Figure 5, compared to other methods, DirectEdit achieves an optimal balance between editability and the preservation of non-edited regions.

**Qualitative Comparison on Real Image Editing.** Based on the PIE-Bench (Ju et al., 2023), we conduct a qualitative comparison of existing image editing methods, as shown in Figure 4. We performed extensive experiments on both synthetic and real-world images, covering a wide variety of editing tasks. It can be observed that IP2P (Brooks et al., 2023), RFEdit (Wang et al., 2024), and FireFlow (Deng et al., 2024) exhibit suboptimal performance in localizing editing targets and preserving non-edited regions, often proving prone to over-editing. While FlowEdit (Kulikov et al., 2025) adheres to editing prompts, it significantly compromises the structural integrity and fine details of the original object. Conversely, although FTEdit (Xu et al., 2025) and DNAEdit (Xie et al., 2025) preserve the global image structure, they fall short in maintaining background fidelity and consistency with editing prompts, often resulting in under-editing. In comparison, DirectEdit excels across diverse editing scenarios, achieving the best balance between preserving background and features while maintaining high alignment with the target prompt. More editing results and

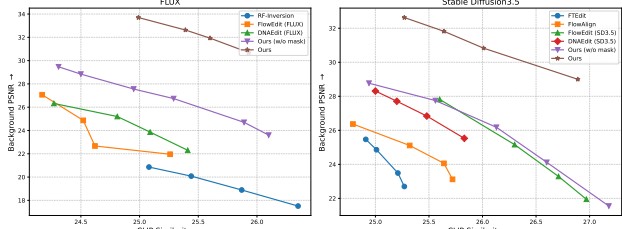

*Figure 5.* **Trade-off between CLIP similarity versus PSNR.** DirectEdit achieves the optimal balance between editability (CLIP) and background preservation (PSNR) compared to other methods. Connected markers represent different hyperparameters.

qualitative comparisons are provided in Appendix E.

**Reconstruction Results.** In Figure 6, we present a comparison of various RF-based methods by evaluating the step-level reconstruction error. The methods compared include Euler Inversion, Stepwise Correction (Ju et al., 2023), RFEdit (Wang et al., 2024), and FTEdit (Xu et al., 2025), all of which are implemented based on FLUX.1-dev. Euler Inversion exhibits the most substantial step-level error, which can be attributed to the amplification effect caused by significant error accumulation. RFEdit (Wang et al., 2024) employs high-order solvers to reduce step-level error but still suffers from severe error accumulation. Stepwise Correction (Ju et al., 2023) employs a strategy that realigns the trajectory with the correct inversion path after each reconstruction step; however, step-level errors remain pronounced. Building on this, FTEdit (Xu et al., 2025) iteratively optimizes the inversion path, which marginally mitigates the reconstruction error, yet the error remains significant during the early and late stages. Consequently, due to the persistent nature of step-level errors, these existing editing methods struggle to preserve the authentic features of images. In con-

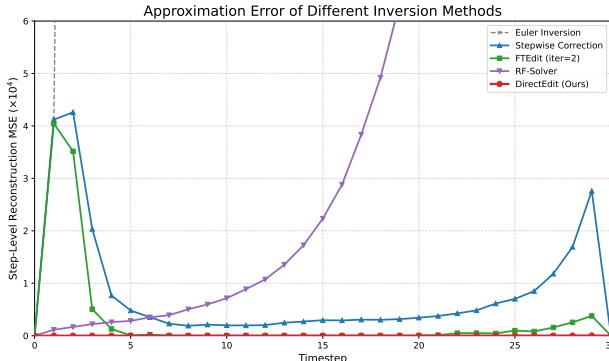

Figure 6. **Comparison of reconstruction errors across different inversion methods.** DirectEdit demonstrates the lowest reconstruction error among all compared approaches.

trast, by aligning the reconstruction path with the inversion trajectory via residuals, DirectEdit achieves step-level accurate reconstruction, thereby surpassing existing methods.

We further present a quantitative comparison of reconstruction errors on PIE-Bench in Table 2. Specifically, in addition to measuring the discrepancy between the reconstructed and source images, we calculate the MSE between the reconstructed and inverted latents after each denoising step, reporting both the average and maximum errors over the entire process. As observed, DirectEdit exhibits superior reconstruction performance across multiple metrics—limited only by the inherent VAE reconstruction loss—without introducing additional NFEs. Crucially, DirectEdit significantly outperforms existing methods in terms of step-level MSE, thereby ensuring more reliable feature interaction.

### 4.3. Ablation and Analysis

We conduct ablation studies on FLUX.1-dev to investigate the impact of various design choices on editing task performance. Specifically, we evaluate the following configurations: (1) Editing results utilizing the Vanilla standard Euler inversion; (2) Without Direct Alignment (corresponding to the *Stepwise Correction* introduced in Section 3.1); (3) Without Attention Injection; (4) Without Mask-guided Blending; and (5) Without the Multi-branch Mask (substituting with rectangular bounding boxes).

Table 4 demonstrates that Direct Alignment significantly enhances structural preservation, background fidelity, and text alignment. When the inversion method is replaced solely with Stepwise Correction (denoted as *w/o alignment*), a marked decline is observed across all metrics, thereby validating the criticality of step-level accurate reconstruction. Attention injection is pivotal for preserving source image details; although removing it slightly improves text alignment scores, it loses fine-grained details of the original object. Finally, the mask-guided blending mechanism

Table 2. **Quantitative comparison of reconstruction errors.** DirectEdit achieves step-level accurate reconstruction without introducing additional NFEs.

| Method | Efficiency | Background Preservation | | | | Step-Level MSE | |
|---|---|---|---|---|---|---|---|
| | NFE ↓ | PSNR ↑ | LPIPS ↓ | MSE ↓ | SSIM↑ | Avg. ↓ | Max ↓ |
| VAE | - | 34.38 | 10.65 | 6.19 | 94.58 | - | - |
| Vanilla | 60 | 14.59 | 421.97 | 466.60 | 50.86 | 1177.7316 | 39511.7188 |
| Stepwise Correction | 60 | 34.38 | 10.65 | 6.19 | 94.58 | 0.2857 | 11.7302 |
| FTEdit | 120 | 34.38 | 10.65 | 6.19 | 94.58 | 0.0881 | 14.8201 |
| RFEdit | 120 | 21.92 | 186.32 | 119.31 | 74.95 | 231.7207 | 20156.2500 |
| DirectEdit (Ours) | 60 | 34.38 | 10.65 | 6.19 | 94.58 | 0.0006 | 0.0757 |

Table 3. **Ablation study on key components**: (1) Direct Alignment, (2) Attention injection, (3) Mask-guided blending, (4) Multibranch mask. Full configuration achieves optimal balance between edit quality and background preservation.

| Component | Structure | Background Preservation | | | | CLIP Similarity | |
|---|---|---|---|---|---|---|---|
| | Distance ↓ | PSNR ↑ | LPIPS ↓ | MSE ↓ | SSIM↑ | Whole ↑ | Edited ↑ |
| Vanilla | 75.95 | 16.81 | 276.29 | 332.65 | 66.54 | 23.57 | 21.79 |
| w/o alignment | 29.22 | 31.12 | 53.16 | 48.17 | 91.29 | 25.24 | 22.35 |
| w/o attention | 23.75 | 31.93 | 39.60 | 33.97 | 92.91 | 25.60 | 22.69 |
| w/o mask | 21.93 | 24.70 | 102.92 | 56.76 | 85.76 | 25.89 | 22.74 |
| w/o multi-branch | 19.15 | 27.86 | 60.92 | 38.94 | 90.43 | 25.71 | 22.70 |
| DirectEdit (Ours) | 17.94 | 32.63 | 35.45 | 25.05 | 93.49 | 25.39 | 22.45 |

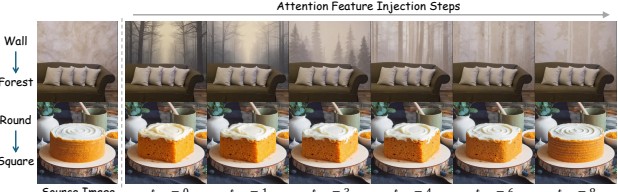

Figure 7. **Ablation study on attention feature injection steps in DirectEdit.** Attention feature injection primarily influences the consistency between the edited region and the source image, where a greater number of injection steps results in higher similarity.

significantly influences background fidelity, and the introduction of multi-branch masks further enhances background preservation. Further ablation study results under various configurations are detailed in Appendix E.2.

We further illustrate the impact of the attention feature injection step $t_{inj}$ in Figure 7. It can be observed that smaller injection steps yield results that align more closely with the target prompt, whereas larger steps result in the edited image bearing greater similarity to the source image. Empirical results indicate that setting the injection step to 3 satisfies the requirements of most editing scenarios. However, in practical applications, this parameter can be adjusted according to specific editing tasks to achieve optimal outcomes.

### 4.4. Limitation

Similar to other training-free editing methods, the editing quality achieved by DirectEdit heavily depends on the prior of the foundation T2I backbone. Furthermore, DirectEdit encounters difficulties in handling specific editing tasks such as size changes, spatial manipulations, and complex

contextual reasoning. A detailed analysis of failure cases is provided in Appendix E.4.

## 5. Conclusion

In this paper, we present DirectEdit, a simple yet highly effective training-free framework for flow-based image editing. Unlike existing methods that strive to rectify the inversion trajectory, DirectEdit enforces explicit alignment between the reconstruction and inversion paths. This strategy effectively eliminates step-level reconstruction errors, thereby ensuring reliable source feature injection. Furthermore, by incorporating multi-branch mask-guided noise blending and attention feature injection, DirectEdit achieves an optimal balance between text-image consistency and background preservation. Both qualitative and quantitative experimental results demonstrate that the proposed DirectEdit exhibits superior performance across diverse editing tasks, enabling high-quality training-free image editing.

## Acknowledgements

This work was supported by the National Key Research and Development Program of China under Grant (2026YFE0202100), the National Natural Science Foundation of China under Grant (T2541022 and 62361166629), and the Major Project of Science and Technology Innovation of Hubei Province under Grant (2025BEA002). The numerical calculations were supported by the supercomputing system at the Supercomputing Center of Wuhan University.

## Impact Statement

The image editing methodology proposed in this work entails several potential societal impacts, encompassing both positive contributions and negative concerns. On the positive side, this method serves as a robust content creation tool for domains such as digital media, artistic design, and scientific research, thereby fostering the advancement of creative industries. Conversely, the potential risks associated with this technology warrant careful consideration. Reliance on web-scraped training data may exacerbate inherent societal biases. Furthermore, there exists a risk of generating misleading content through the manipulation of human portraits. To mitigate these concerns, the establishment of regulatory guidelines and robust ethical frameworks is imperative.

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

## A. More Details of the DirectEdit Algorithm

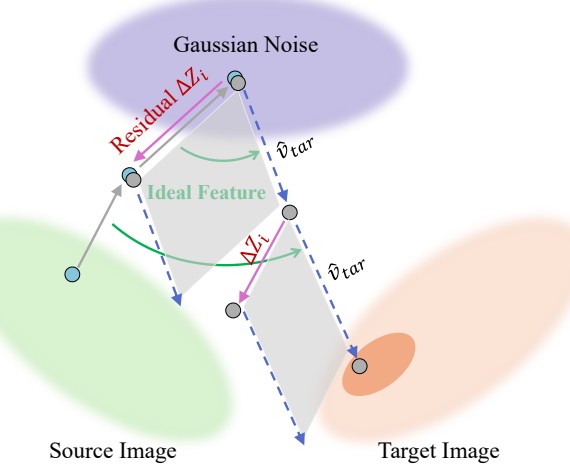

*Figure 8.* DirectEdit with Virtual Reconstruction.

As shown in Algorithm 1, DirectEdit achieves precise reconstruction and facilitates drift-free feature interaction. Crucially, we observe that the editing process leverages features extracted from the reconstruction path. Given that the reconstruction path is explicitly aligned with the inversion trajectory in our framework, the computation of the reconstruction path effectively becomes redundant for the sole purpose of feature extraction. Consequently, if the primary objective is strictly the editing task, the explicit reconstruction path can be eliminated, as illustrated in Figure 8.

Building on this insight, we further propose **DirectEdit with Virtual Reconstruction**. This variant eliminates the reconstruction path by directly interacting the latent features from the inversion process with the editing path. This modification enhances algorithmic efficiency but at the cost of increased memory usage for feature storage, as detailed in Algorithm 2. It is crucial to emphasize that the alignment operation denoted as $\hat{\mathbf{Z}}_i^{tar} \leftarrow \mathbf{Z}_i^{tar} + \Delta\mathbf{Z}_i$ is indispensable. While prior works (Zhu et al., 2025; Wang et al., 2024) have similarly employed feature caching techniques, they notably lack this explicit alignment, thereby injecting drifted features. This operation ensures that the computation of the velocity field $v_t^{tar}$ is performed on latents that are strictly noise-aligned with the virtual reconstruction path. Consequently, this guarantees the consistency of feature interactions between the paths. In the absence of such alignment, discrepancies between latent representations may lead to unexpected semantic drift, thereby degrading image fidelity.

---

**Algorithm 2** DirectEdit with Virtual Reconstruction

---

**Input:** Source image $\mathbf{I}_{src}$, Source text $\psi_{src}$, Target text $\psi_{tar}$, Timesteps $\{\sigma_0, \ldots, \sigma_T\}$, Denoising steps $T$.

$\mathbf{Z}_T \leftarrow \text{VAE\_Encode}(\mathbf{I}_{src})$          ▷ *Inversion Process*

**for** $t = T-1$ **to** $0$ **do**

    $\mathbf{Z}_t^{inv} \leftarrow \mathbf{Z}_{t+1}^{inv} - (\sigma_{t+1} - \sigma_t) v_\theta(\mathbf{Z}_{t+1}^{inv}, \psi_{src})$      ▷ *Save attention features for editing*

    $\Delta\mathbf{Z}_t \leftarrow \mathbf{Z}_{t+1}^{inv} - \mathbf{Z}_t^{inv}$          ▷ *Save latent residuals*

**end for**

$\mathbf{Z}_0^{tar} \leftarrow \mathbf{Z}_0^{inv}$          ▷ *Editing Process*

$O, P, Q \leftarrow \text{MLLM}(\mathbf{I}_{src}, \psi_{src}, \psi_{tar})$

$\mathcal{M} \leftarrow \mathcal{M}(O, P, Q)$          ▷ *Generate multi-branch mask*

**for** $t = 0$ **to** $T-1$ **do**

    $\hat{\mathbf{Z}}_t^{tar} \leftarrow \mathbf{Z}_t^{tar} + \Delta\mathbf{Z}_t$          ▷ *Direct Alignment*

    $\hat{v}_t^{tar} \leftarrow v_\theta(\hat{\mathbf{Z}}_t^{tar}, \psi_{tar})\{\mathbf{F}_t^{tar} \leftarrow \hat{\mathbf{F}}_t^{tar}\}$      ▷ *Inject attention features*

    $\mathbf{Z}_{t+1}^{tar} \leftarrow \mathbf{Z}_t^{tar} + (\sigma_{t+1} - \sigma_t)\hat{v}_t^{tar}$

    $\mathbf{Z}_{t+1}^{tar} \leftarrow \mathbf{Z}_{t+1}^{inv} \odot (\mathbf{1} - \mathcal{M}) + \mathbf{Z}_{t+1}^{tar} \odot \mathcal{M}$      ▷ *Mask-guided blending*

**end for**

**Output:** $\text{VAE\_Decode}(\mathbf{Z}_T^{tar})$

---

## B. Comparison with Null-text Inversion

DirectEdit shares similar inversion objectives with some existing methods. For instance, Null-text Inversion (Mokady et al., 2023) stands out as a prominent test-time optimization technique within the realm of diffusion models. Conceptually analogous to DirectEdit, this method regards the DDIM inversion trajectory as a pivot. It proceeds by optimizing a distinct null-text embedding $\{\emptyset_t\}_{t=1}^T$ for each timestamp $t$ during the forward process, utilizing the embedding from the previous step, $\emptyset_{t+1}$, to initialize $\emptyset_t$. By treating the null-text embedding as a learnable parameter, it optimizes the specific $\emptyset_t$ corresponding to each timestep during the sampling phase, as formulated in Equation (15):

$$\min_{\emptyset_t} \left\| z_{t-1}' - z_{t-1}(z_t, \emptyset_t, C) \right\|_2^2 \tag{15}$$

where $z_{t-1}(z_t, \emptyset_t, C)$ denotes the application of a DDIM sampling step utilizing the latent $z_t$, the unconditional embedding $\emptyset_t$, and the conditional embedding $C$. Subsequently, the real input image can be edited employing the starting noise $z_T = z_T'$ and the sequence of optimized unconditional embeddings $\{\emptyset_t\}_{t=1}^T$.

Fundamentally, the optimization objective of Null-text Inversion is equivalent to Equation (7). The optimized null-text embeddings influence both the reconstruction and editing paths, playing a role analogous to the latent residual $\Delta \mathbf{Z}_t$ in our DirectEdit algorithm. In comparison, however, DirectEdit achieves superior fidelity in both reconstruction and editing while entirely eliminating the need for computationally expensive test-time optimization.

## C. Comparison with DNAEdit

DNAEdit (Xie et al., 2025) represents a state-of-the-art inversion method based on flow models, which employs a residual offset mechanism analogous to ours to facilitate inversion and editing. Specifically, leveraging the linearity inherent in Rectified Flow (RF) during the inversion process, DNAEdit constructs a linear trajectory from the noise $S_{t+1}$ to the latent variable $Z_{t+1}$. It then estimates the latent variable $Z_t$ via interpolation, denoted as $Z_t^*$:

$$Z_t^* = \frac{\sigma_t}{\sigma_{t+1}} \times Z_{t+1} + \left(1 - \frac{\sigma_t}{\sigma_{t+1}}\right) \times S_{t+1}. \tag{16}$$

Using this estimated noisy latent $Z_t^*$, the velocity is predicted as $v_t^{src} = v_\theta(Z_t^*, \psi^{src})$. The method calculates the velocity difference $\Delta v_t^{\text{DNA}} = v_t^{\text{linear}} - v_t^{src}$ to update both $Z_t = Z_t^* + \Delta v_t^{\text{DNA}} \times (\sigma_{t+1} - \sigma_t)$ and the Gaussian noise used for interpolation, while concurrently recording the residual offset $\Delta x_t^{\text{DNA}} = Z_t^* - Z_t$. During the editing phase, the noisy latent $Z_t^{\text{edit}}$ is first combined with the residual $\Delta x_t^{\text{DNA}}$ to align with the inversion process, followed by the prediction of the update velocity $v_t^{tgt}$:

$$Z_t^{*\text{edit}} = Z_t^{\text{edit}} + \Delta x_t^{\text{DNA}}, \quad v_t^{tgt} = v_\theta(Z_t^{*\text{edit}}, \psi^{tgt}). \tag{17}$$

Note that there exists a trajectory $Z_t^{\text{mvg}}$ transitioning from the source image to the target image entirely within the image space. To preserve source image features, DNAEdit introduces a technique termed Mobile Velocity Guidance (MVG). Prior to updating the noisy latent $Z_t^{\text{edit}}$, the trajectory $Z_t^{\text{mvg}}$ is shifted from the source image direction towards the target image direction by applying the velocity difference:

$$\Delta v_t = v_t^{\text{tgt}} - v_t^{\text{src}}, \quad Z_{t+1}^{\text{mvg}} = Z_t^{\text{mvg}} + \Delta v_t \times (\sigma_{t+1} - \sigma_t). \tag{18}$$

By blending the velocity $v_t^{\text{tgt}}$, which is based entirely on the target text, with the MVG velocity $v_t^{\text{mvg}}$ using a weighting coefficient $\eta$, the final denoising velocity is obtained:

$$v_t^{\text{mvg}} = \frac{Z_t^{\text{edit}} - Z_{t+1}^{\text{mvg}}}{1 - \sigma_t}, \quad v_t^{\text{edit}} = \eta \times v_t^{\text{tgt}} + (1 - \eta) \times v_t^{\text{mvg}}. \tag{19}$$

Finally, denoising is performed using the synthesized velocity: $Z_{t+1}^{\text{edit}} = Z_t^{\text{edit}} + v_t^{\text{edit}}(\sigma_{t+1} - \sigma_t)$. However, due to the reliance on linear interpolation, DNAEdit aligns with a linear estimate rather than the actual inversion trajectory, thereby introducing errors and feature drift. In contrast, DirectEdit aligns directly with and reconstructs the actual inversion trajectory, ensuring both efficiency and the consistency of latent features. Furthermore, our empirical experiments indicate that utilizing attention features from the source branch yields greater stability and flexibility than the velocity blending employed in MVG. When combined with our proposed noise blending module, our approach achieves superior background preservation.

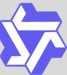

**System Prompt:**
"""
You are an intelligent image editing assistant.
Analyze the source image, source prompt, and target prompt to determine the editing task and the region of interest.
**Task Definitions:**
1. Local: Editing a specific object or its attributes.
   Examples: Replacing a dog with a cat, removing a person, changing a shirt's color to red, altering facial expression.
2. Background: Changing the environment or background details while preserving the main subject.
   Examples: Changing the setting from a street to a beach, changing the background color to white, moving the subject to a forest.
3. Global: Holistic changes affecting the entire image atmosphere or style.
   Examples: Turning a photo into an oil painting, changing summer to winter, converting day to night, cyberpunk style transfer.
4. Other: Structural changes, additions, or partial modifications in a specific non-object region.
   Examples: Adding a bird in the sky, adding glasses to a face, modifying a specific texture patch.
**Output a JSON object strictly with these fields:**
- "type": One of ["Local", "Background", "Global", "Other"].
- "bbox": [x1, y1, x2, y2] representing the bounding box of the region of interest using normalized coordinates (0-1000).
  CRITICAL BBOX RULES based on "type":
  - For "Local": Enclose the SPECIFIC OBJECT that needs to be edited, removed, or altered.
  - For "Background": Enclose the FOREGROUND OBJECT (the subject) that must be PROTECTED/PRESERVED.
  - For "Other": Enclose the target region where the new content will be added or modified.
  - For "Global": Return [0, 0, 1000, 1000].
"""

**User:**

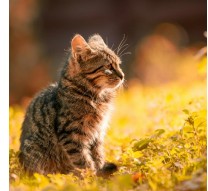

**Source Prompt:** A small, fluffy kitten sitting in a grassy field. The kitten is positioned in the center of the scene, surrounded by a field.
**Target Prompt:** A small, fluffy puppy sitting in a grassy field. The puppy is positioned in the center of the scene, surrounded by a field.

**Response:**

```
{
  "type": "Local",
  "bbox": [48, 179, 602, 903]
}
```

*Figure 9.* System prompt for generating multi-branch mask.

## D. More Implementation Details

In our experiments, we implement our method utilizing FLUX.1-dev (Labs, 2024) and SD3.5-medium (Esser et al., 2024) as the backbone models, respectively. For both architectures, the number of denoising steps is uniformly set to 30. We configure the Classifier-Free Guidance (CFG) (Ho & Salimans, 2022) scale to 1 for the inversion process and 2 for the editing process. To mitigate the impact of minor error sources beyond approximation errors, we also employ the stepwise correction strategy introduced in Section 3.1. All results presented in this paper are sourced from PIE-Bench (Ju et al., 2023) and royalty-free online resources (Pexels, 2024). We benchmark DirectEdit against prior editing approaches (Brooks et al., 2023; Hertz et al., 2022; Wang et al., 2024; Deng et al., 2024; Kulikov et al., 2025; Xu et al., 2025; Xie et al., 2025; Kim et al., 2025), with all baselines evaluated using their officially recommended hyperparameters. We employ Qwen3-VL (Yang et al., 2025a) as the Multimodal Large Language Model (MLLM) within the multi-branch mask-guided blending module. The specific system prompt used is illustrated in Figure 9. To mitigate potential edge misalignment issues associated with segmentation masks generated by SAM (Kirillov et al., 2023), we apply a minor morphological dilation (kernel size = 5) to smooth the mask boundaries. Regarding the attention injection settings, we set the injection timestep threshold $t_{inj}$ to 3. Specifically, for SD3.5 (Esser et al., 2024), we inject Value features into all MM-DiT blocks with the exception of the final one. Conversely, for FLUX (Labs, 2024), we apply feature injection exclusively to all single-stream blocks.

## E. Additional Results

### E.1. More Comparisons on PIE-Bench

Figure 10 provides additional qualitative comparisons on PIE-Bench, where the left panel displays results from Stable Diffusion-based methods and the right panel presents those from FLUX-based methods. Our method consistently exhibits superior editing and preservation performance across diverse tasks.

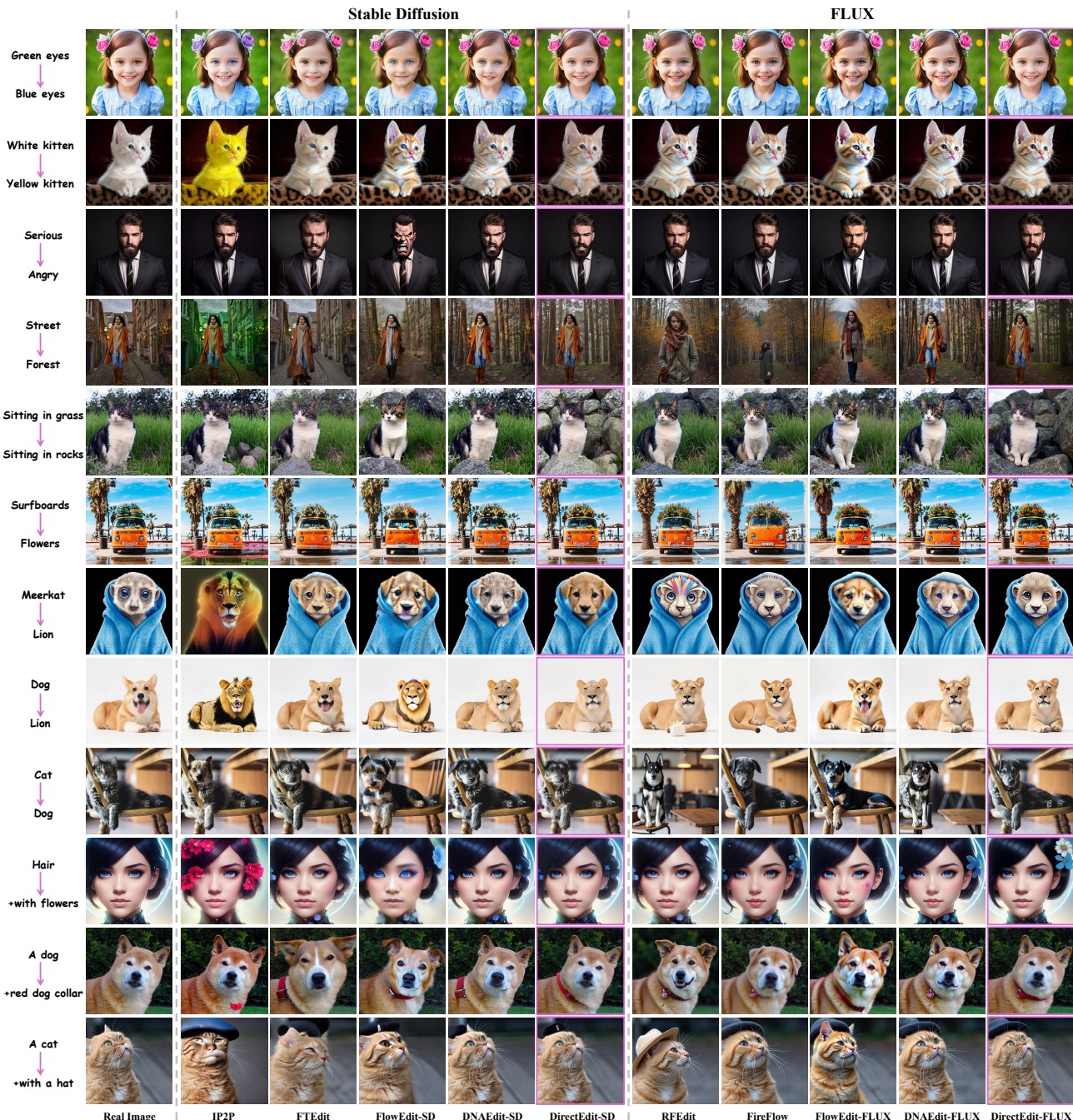

*Figure 10.* Additional qualitative comparisons on PIE-Bench.

## E.2. More Ablation Study

We conducted additional ablation studies on FLUX.1-dev to investigate the impact of various design choices on editing performance. Exp. ① employ the standard Euler method; due to the absence of editing components such as attention injection and noise blending, this configuration is equivalent to direct generation guided by the target prompt. Consequently, it performs poorly across all preservation metrics, despite achieving high CLIP similarity. Comparing Exp. ③, ④ and ⑨, we observe that Exp. ⑨ outperforms Exp. ③ (Euler) and Exp. ④ (Stepwise Correction) across all metrics, thereby demonstrating the effectiveness of our proposed Direct Alignment. Furthermore, a comparison between Exp. ⑤ and Exp. ⑨ reveals that attention injection plays a pivotal role in preserving the details of the source image. In Exp. ⑦, Direct

*Table 4.* **Ablation study on various settings**. Full configuration achieves an optimal balance between editability and background fidelity.

| Exp. | Component | Structure Distance ↓ | Background Preservation PSNR ↑ | LPIPS ↓ | MSE ↓ | SSIM↑ | CLIP Similarity Whole ↑ | Edited ↑ |
|---|---|---|---|---|---|---|---|---|
| ① | Euler | 89.51 | 14.85 | 298.85 | 484.52 | 64.93 | 26.82 | 23.60 |
| ② | Euler+attn | 75.95 | 16.81 | 276.29 | 332.65 | 66.54 | 23.57 | 21.79 |
| ③ | Euler+attn+mask | 74.95 | 16.66 | 276.50 | 338.42 | 66.40 | 23.50 | 21.35 |
| ④ | S.C.+attn+mask | 29.22 | 31.12 | 53.16 | 48.17 | 91.29 | 25.24 | 22.35 |
| ⑤ | Align+mask | 23.75 | 31.93 | 39.60 | 33.97 | 92.91 | 25.60 | 22.69 |
| ⑥ | Align+attn | 21.93 | 24.70 | 102.92 | 56.76 | 85.76 | 25.89 | 22.74 |
| ⑦ | Align (src)+attn | 44.64 | 20.59 | 184.11 | 115.60 | 76.63 | 25.76 | 22.66 |
| ⑧ | Align+attn+mask (rec) | 19.15 | 27.86 | 60.92 | 38.94 | 90.43 | 25.71 | 22.70 |
| ⑨ | Align+attn+mask | 17.94 | 32.63 | 35.45 | 25.05 | 93.49 | 25.39 | 22.45 |

Alignment is applied only to the source branch (i.e., $\hat{\mathbf{Z}}_t^{src} = \mathbf{Z}_t^{src} + \Delta \mathbf{Z}_t, \quad \hat{\mathbf{Z}}_t^{tar} = \mathbf{Z}_t^{tar}$). A comparison with Exp. ⑥ shows that Exp. ⑥ consistently exhibits superior performance across all metrics, verifying the efficacy of our proposed design. Finally, the mask-guided blending mechanism significantly influences background fidelity, and the introduction of multi-branch masking further enhances background preservation (see Exp. ⑥, ⑧, and ⑨).

### E.3. More Results on Real Images

Figure 11 presents additional results of DirectEdit applied to high-resolution real-world images. As observed, DirectEdit achieves versatile image editing across diverse scenarios, capable of handling diverse local edits (e.g., object replacement, attribute modification, and fine-grained editing) as well as global style transfer (e.g., transforming photographs into painting or cartoon styles). Furthermore, by leveraging the advantage of step-level accurate reconstruction, DirectEdit maintains high background fidelity while offering editing flexibility. This enables not only non-rigid editing, such as pose change, but also structural modifications like object addition and removal.

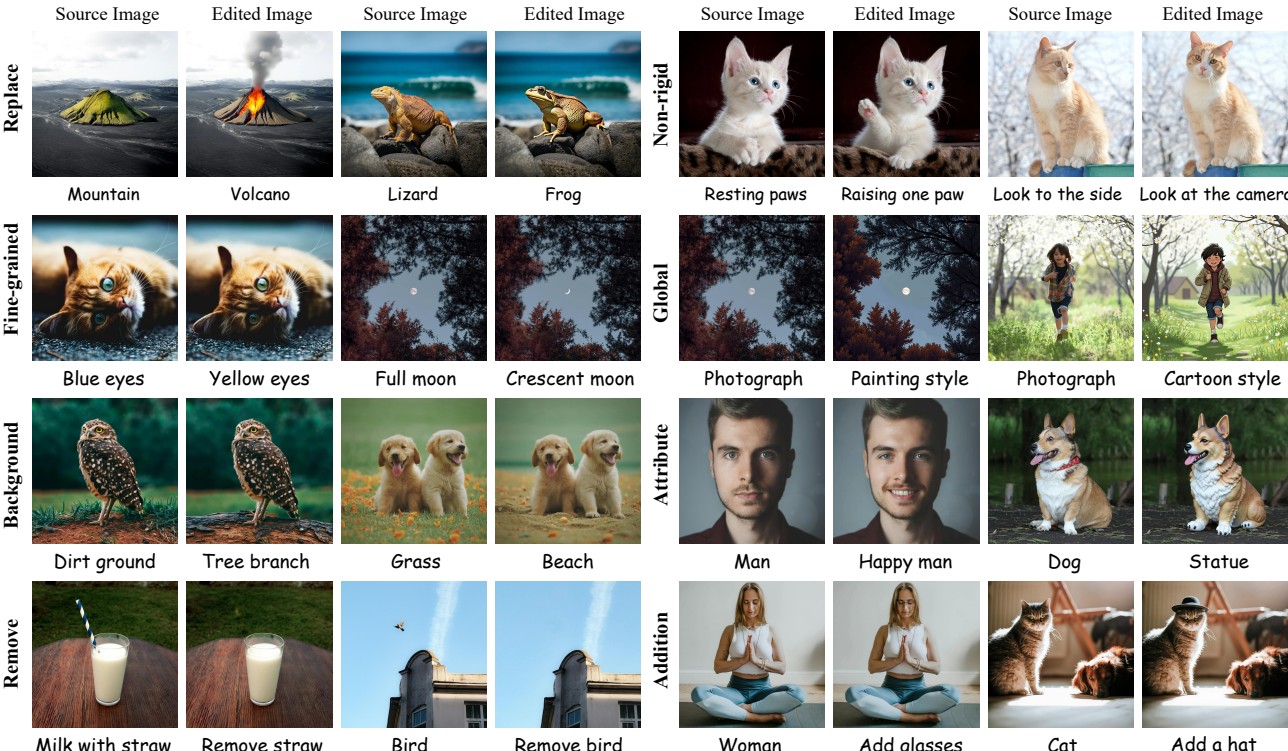

*Figure 11.* Additional editing results on high-resolution real-world images.

## E.4. Failure Case Study

We present failure cases in Figure 12. Although DirectEdit excels across a diverse range of training-free editing tasks, it exhibits limitations in specific scenarios, such as resizing, spatial manipulations, and complex contextual reasoning. We attribute these failures primarily to the inherent limitations of the underlying T2I model and the structural nature of the inverted noise. In future work, we will focus on addressing these aforementioned challenges.

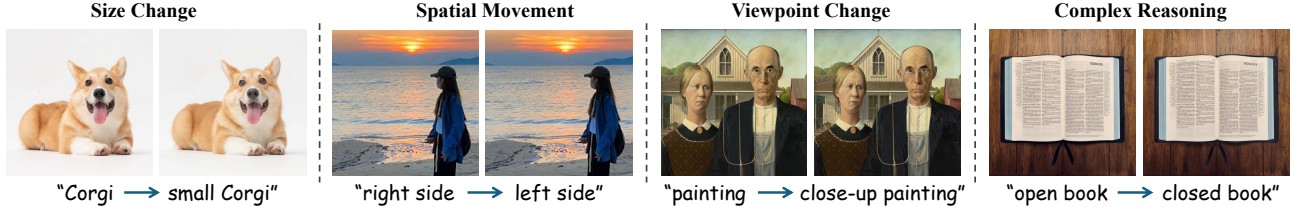

*Figure 12.* Failure case study. Our method shows inherent limitations when handling specific editing tasks (e.g., size change, spatial movement, viewpoint change, complex reasoning).

