# OpenReview forum: "DirectEdit: Step-Level Accurate Inversion for Flow-Based Image Editing"
_ICML.cc/2026/Conference — ICML 2026 regular_

### Official Review · Reviewer_c8wy · 2026-03-05

**Soundness:** 3
**Presentation:** 3
**Significance:** 2
**Originality:** 3
**Overall Recommendation:** 4
**Confidence:** 5

**Summary:**

The author propose a training free image editing solution for image editing and summarize the major contributions as: 1) the first model for flow inversion with step-level accuracy. 2) a new preservation mechanism (I will discuss this later) combining multi-branch mask-guided noise blending with attention-based feature injection. (to be honest, the presentation here is not necessary to be so complication. It would be better to straight forward.) 3) extensive experiments to show SOTA performance.

The math trick of this work is clever and overall the idea to use inversion residual to calibrate forward velocity is novel to me.

**Compliance With Llm Reviewing Policy:**

Affirmed.

**Key Questions For Authors:**

The paper attributes the reconstruction drift primarily to the approximation. From a theoretical standpoint, it would be helpful to clarify whether there exists a formal argument showing that this approximation dominates the error propagation in the inversion dynamics. Otherwise, it is not immediately clear why other potential sources of error in the denoising process would be negligible.

The core contribution of the paper relies on that the inversion trajectory is the GT reference, yet, this is not true in practice. Because the inversion itself is also an approximation. That means you can have a perfect aligned trajectory, while the trajectory is biased: velocity matching ≠ global trajectory correctness.

Overall I like the idea, but the paper would be strengthen by discussing well those limitations.

**Limitations:**

Yes

**Strengths And Weaknesses:**

The paper clearly identifies a key issue in inversion-based diffusion editing, namely the accumulation of approximation errors caused by the velocity substitution. The method is training free, so it applicable to continuous time models in practices. The results looks significant;

Aligning the forward reconstruction path to this approximate inversion trajectory does not necessarily guarantee globally accurate reconstruction, especially when the inversion error becomes large or when the number of sampling steps is small.

---

> ### Author Rebuttal · Authors · 2026-03-31
>
> We sincerely appreciate your time and effort in reviewing our paper. We hope our following point-to-point responses can address your concerns.
>
> **Question 1: Formal argument on approximation dominating the error propagation.**
>
> Thank you for the insightful question. We can formally clarify why the approximation error dominates the error propagation.
>
> **Theoretical Argument.** Within the Euler inversion framework, the primary error sources include: (1) the asymmetric approximation error from using $v_\theta(Z_{t+1}^{inv})$ to approximate $v_\theta(Z_t^{inv})$, (2) the Euler discretization truncation error, and (3) the neural network estimation error ($\epsilon_{net}$). Crucially, these errors have fundamentally different mathematical properties regarding trajectory divergence. The Euler discretization error affects both the inversion and forward processes identically since both are governed by the same ODE with identical step sizes. Similarly, while the pretrained network is imperfect ($v_\theta = v_{true} + \epsilon_{net}$), the exactly identical network is used for both paths. Therefore, the network error acts symmetrically. While $\epsilon_{net}$ affects the overall generation quality (e.g., semantic blurriness), it largely cancels out when computing the relative trajectory divergence (drift) between the inversion and reconstruction paths. In contrast, the approximation error exists *only* in the inversion path, making it a strictly asymmetric driving source of the trajectory mismatch. Due to the Lipschitz continuity of the highly non-linear vector field, this specific local error is exponentially amplified at each step. These small per-step asymmetric errors compound over $T$ steps, acting as the sole dominant force leading to severe reconstruction drift.
>
> **Empirical Validation.** **Table 2** of our paper provides direct evidence supporting this formal analysis. The "Stepwise Correction" baseline structurally eliminates global error accumulation, meaning its Step-Level MSE reflects the combined single-step effect of all error sources. DirectEdit explicitly eliminates *only* the approximation error (1), without modifying the model weights or discretization scheme. As the results show, the Avg. MSE drops massively from **0.2857 to 0.0006** (a **~475× reduction**), and the Max MSE drops from **11.7302 to 0.0757**. This near-zero residual error provides proof that once the asymmetric approximation error is removed, the remaining symmetric errors are indeed negligible regarding reconstruction drift.  We will incorporate these discussion into the final version to strengthen our theoretical foundation.
>
> **Question 2: The inversion bias and global trajectory correctness.**
>
> Thank you for raising this insightful question. In training-free image editing, inversion itself is indeed an approximation. Therefore, the inversion trajectory $Z_{inv}(t)$ has an inherent bias. However, we want to clarify that the core goal of DirectEdit is not to guarantee the absolute correctness of the global trajectory. Instead, it aims to ensure reliable feature exchange. As stated in **Section 3.2** of our paper, we treat the inversion path as a "pivot." This idea is very similar to Null-text Inversion (please see **Appendix B** for a detailed comparison). By doing this, DirectEdit achieves step-level accurate reconstruction. This ensures that the source image features extracted at each step do not drift. Consequently, it guarantees the reliability of the feature exchange and improves the fidelity of the edited images. We will add more discussion on this in the final version.

---

> > ### Author Rebuttal · Reviewer_c8wy · 2026-04-03
> >
> > Thanks! my concerns are addressed, so I will keep my positive score.

---

### Official Review · Reviewer_mnnY · 2026-03-12

**Soundness:** 3
**Presentation:** 4
**Significance:** 4
**Originality:** 3
**Overall Recommendation:** 5
**Confidence:** 3

**Summary:**

This paper proposes an accurate step‑level inversion method based on rectified flow for training‑free image editing. The method avoids the accumulated errors present in existing Euler‑based and correction‑based inversion approaches, resulting in accurate alignment between the inversion and reconstruction trajectories. The proposed inversion technique is supported by both mathematical analysis and empirical validation. In addition, the authors introduce a multi‑branch, mask‑guided noise‑blending strategy and an attention‑based feature‑injection mechanism to balance editing fidelity and editability. Extensive experiments based on Stable Diffusion and FLUX demonstrate impressive inversion accuracy and editing performance.

**Compliance With Llm Reviewing Policy:**

Affirmed.

**Final Justification:**

Please see the Rebuttal Acknowledgement

**Key Questions For Authors:**

1. As described in Section 3.3, the blending mask is generated using an MLLM and SAM. It would strengthen the paper to analyze the robustness of the method with respect to imperfect masks. For instance, if the mask misses parts of the target object, to what extent would the editing results degrade?
2. In Table 3, since the accurate inversion method is a key contribution of this work, adding an ablation study that evaluates the editing performance using only the proposed inversion component would help clarify its direct impact on the overall framework.

**Limitations:**

yes

**Strengths And Weaknesses:**

Strengths:
1. The paper is well written, with a clearly articulated motivation and a coherent narrative supported by effective illustrations and explanations.
2. The proposed method achieves accurate step‑level inversion and substantially mitigates the error‑accumulation issues present in existing approaches, which is crucial for training‑free image editing.
3. The accuracy of the step‑level inversion is convincingly validated through both mathematical analysis and empirical experiments.
4. The method delivers strong editing performance, demonstrating its practical effectiveness across various scenarios.

Weaknesses:
1. Although Table 2 provides quantitative evidence of inversion accuracy, it would be beneficial to also include visual comparisons of reconstruction results against existing methods. Such qualitative results would more intuitively highlight the superiority of the proposed inversion scheme.
2. The editing modes showcased in Figure 4 appear relatively simple. Including more challenging editing tasks, such as significant pose changes, would better demonstrate the robustness and versatility of the proposed model.

---

> ### Author Rebuttal · Authors · 2026-03-31
>
> We sincerely appreciate your time and effort in reviewing our paper. We hope our following point-to-point responses can address your concerns.
>
> **Weakness 1: Visual comparisons of reconstruction results.**
>
> Thank you for carefully reading our paper and for the suggestion. We provide visual comparisons of the reconstruction results against existing methods the following [anonymous link](https://ibb.co/27M2wb4y). However, we want to highlight a key point discussed in Sections 3.1 and 4.2. Some methods (like FTEdit) use a "stepwise correction" strategy. They manually reset the reconstruction latent to the inversion latent ($Z_t^{src} = Z_t^{inv}$) after every denoising step. Therefore, these methods inevitably produce artificially perfect final visual reconstructions. Visual comparisons of the final reconstructions simply cannot show this step-level error. In contrast, DirectEdit achieves accurate step-level reconstruction which significantly improves the fidelity of the edited images. We will add more discussion on this in the final version.
>
> **Weakness 2: Evaluation on more challenging editing tasks.**
>
> We sincerely thank the reviewer for this constructive suggestion. To further prove our method's superiority and versatility on more complex professional editing tasks, we conducted an extra quantitative comparison. We used a high-resolution, 100-image subset of the GIE-Bench dataset. As shown in the following [anonymous link](https://ibb.co/R4V7WvTQ), we added extra metrics like MCQ-based functional correctness. The results show that DirectEdit clearly outperforms existing methods. In addition, we kindly ask the reviewer to check **Appendix E (Figure 10)** in our paper. This section shows our framework's ability to handle high-resolution real-world images. It includes examples of significant pose changes and non-rigid transformations (for example, modifying an animal's pose or angle). We will add more discussion on this in the final version.
>
> **Question 1: Robustness with respect to imperfect masks.**
>
> Thank you for the suggestion. The mask generation pipeline works well in most cases. However, for objects that are hard to segment accurately (like fluffy animals), it might cause unintended edge preservation. To actively fix this issue and improve robustness, we apply a minor morphological dilation operation (mentioned in **Appendix D** of our paper). This simple step effectively avoids unintended edge preservation and ensures smooth semantic transitions ([link](https://ibb.co/pjhN5Brp)). We will add this study and more discussion in the final version.
>
> **Question 2: Ablation study using only the proposed inversion component.**
>
> Thank you for the insightful suggestion. To show the direct impact of our accurate inversion, we kindly ask the reviewer to check the **"w/o blending"** setting in **Table 3** of our paper. This setting completely disables the mask-guided blending and shows the true impact of our proposed inversion component on the overall framework. In addition, we provide a direct quantitative comparison on PIE-Bench in the following [anonymous link](https://ibb.co/vCJHZs8k), which compares DirectEdit "without MLLM/SAM" against all baseline models. As shown in the table, even when using only the proposed inversion method, DirectEdit still outperforms the baselines. We will add this study in the final version.

---

> > ### Author Rebuttal · Reviewer_mnnY · 2026-04-01
> >
> > The reviewer thanks the authors for their detailed response and results. After reading the response, my concerns are adequately addressed. Therefore, I will keep my positive rating.

---

> > > ### Author Response · Authors · 2026-04-01
> > >
> > > We are very glad that our response has adequately addressed this reviewer's concerns. We sincerely thank you for your constructive suggestions and your positive support for our work.
> > >
> > > Authors of paper #10252

---

### Official Review · Reviewer_RCwP · 2026-03-12

**Soundness:** 2
**Presentation:** 3
**Significance:** 3
**Originality:** 2
**Overall Recommendation:** 4
**Confidence:** 4

**Summary:**

This paper presents DirectEdit, a training-free method for inversion-based image editing. To overcome the accumulated error drift inherent in standard Euler inversion, DirectEdit introduces "Direct Alignment," which explicitly aligns the forward reconstruction path with the inversion trajectory step-by-step using latent residuals . Additionally, it utilizes attention feature injection and a multi-branch mask-guided noise blending mechanism during the forward process to ensure precise background preservation and balance fidelity with editability.

**Compliance With Llm Reviewing Policy:**

Affirmed.

**Final Justification:**

My concerns have been well addressed in the rebuttal, thus I raise my score.

**Key Questions For Authors:**

see weakness

**Limitations:**

yes

**Strengths And Weaknesses:**

**Strengths**

1. DirectEdit is a training-free method that achieves Step-Level Accurate Reconstruction
2. Attention feature injection and multi-branch mask-guided noise blending mechanism improve the background preservation and balance fidelity with editability

**Weaknesses**

1. The authors claim that DirectEdit is 'simple yet **efficient**'; however, this claimed efficiency is not justified or discussed later in the paper. If the authors are implying that the method is efficient because it does not introduce additional NFEs compared to FTEdit and RFEdit, it should be noted that many existing methods (such as FlowEdit, RF-Inversion, and FlowAlign) also avoid introducing extra NFEs and DirectEdit doesn't decrease the NFEs compared to vanilla method. Furthermore, relying on SAM to generate masks and Qwen to obtain bounding boxes introduces additional processing steps and computational overhead. Therefore, rather than improving efficiency, the proposed pipeline actually appears to decrease it. The authors should clarify their claim regarding efficiency

2. Sections 3.1 and 3.2 provide a theoretical derivation for Direct Alignment ($\hat{Z}_t^{tar} = Z_t^{tar} + \Delta Z_t$ in Equation (9)) , demonstrating how adding the latent residual ($\Delta Z_t$) achieves step-level accurate **reconstruction** by eliminating approximation errors.  The paper fails to explain why applying the source trajectory's residual to the **newly generated target** trajectory is **mathematically valid**.

3. In the ablation study, the "w/o alignment" setting is implemented by replacing the proposed Direct Alignment ($\hat{Z}_t^{tar} = Z_t^{tar} + \Delta Z_t$) with "Stepwise Correction" (As described in Section 3.1, Stepwise Correction enforces $Z_t = Z_t^{inv}$ during the forward process). A rigorous ablation should simply disable the residual addition (i.e., using $\hat{Z}_t^{tar} = Z_t^{tar}$) to isolate its true effect.

4. While the authors claim that DirectEdit achieves an optimal balance between editability and the preservation of non-edited regions , the quantitative results in Table 1 reveal a persistent trade-off that is difficult to evaluate intuitively through tabular data alone. For example, under the FLUX backbone, FireFlow achieves higher CLIP Similarity scores than DirectEdit, albeit with worse Background Preservation. Similarly, using the SD3.5 backbone, FlowEdit outperforms DirectEdit in Edited CLIP Similarity but falls short in background metrics.

   the "Avg. Rank" metric presented in Table 1 is structurally imbalanced. The average is calculated across five metrics dedicated to structural and background fidelity (Distance, PSNR, LPIPS, MSE, SSIM) but only two metrics dedicated to editability (CLIP Similarity Whole and Edited). To make the claim of a "superior balance" significantly more convincing, it is highly recommended to include a 2D scatter plot (similar to the evaluations in *FlowEdit* or *FlowAlign*)， which will provide an immediate, intuitive visual confirmation of DirectEdit's overall superiority and its exact position on the trade-off frontier.

5. Lack of **User Study** and Comparison with **FlowAlign**[1].

[1] FlowAlign: Trajectory-Regularized, Inversion-Free Flow-based Image Editing. 2025

---

> ### Author Rebuttal · Authors · 2026-03-31
>
> We sincerely appreciate your time and effort in reviewing our paper. We hope our following point-to-point responses can address your concerns.
>
> **Weakness 1: Efficiency claims and MLLM/SAM Overhead.**
>
> We thank you for carefully reading our paper and pointing out this issue. We want to clarify that "simple yet efficient" mainly refers to the optimal balance between generation quality and inference speed, rather than just pure speed. Although some methods also avoid extra NFEs, their persistent errors severely degrade editing performance.
>
> **Regarding the MLLM/SAM Overhead:** The MLLM/SAM mask generation pipeline introduces very little computational overhead (only about 1.3 seconds per image on average). The generated masks can be easily saved and reused across multiple editing turns. Importantly, even when the MLLM/SAM pipeline is completely disabled, DirectEdit still significantly outperforms existing baselines. We provide a direct quantitative comparison on PIE-Bench in the following [anonymous link](https://ibb.co/vCJHZs8k), which compares DirectEdit "without MLLM/SAM" against all baselines. We will add more discussion on this in the final version.
>
> **Weakness 2: Mathematical validity of applying source residual to the target trajectory.**
>
> Thank you for raising this concern. In Rectified Flow (RF), $Z_t$ can be constructed via linear interpolation between the clean image and Gaussian noise :
> $Z_t = (1 - \sigma_t) X_0 + \sigma_t X_1,$ $Z_{t+1} = (1 - \sigma_{t+1}) X_0 + \sigma_{t+1} X_1.$ According to the formula, the residual $\Delta Z_t = Z_{t+1} - Z_t$ is equivalent to the weighted difference between two Gaussian noise samples and therefore free from image content. Explicitly applying the source residual to the target trajectory maintains consistency between the two trajectories and guarantees the accuracy of shared features. Furthermore, this cross-trajectory application of residuals has a strong theoretical foundation in diffusion editing. We kindly ask the reviewer to refer to **Appendix B and C** of our paper for more details.
>
> **Weakness 3: Rationale for the "w/o alignment" ablation setting.**
>
> **Why we used "Stepwise Correction" as the baseline:** Specifically, "Direct Alignment" works on both the source path and the target path ($\hat{Z}_t^{src} = Z_t^{src} + \Delta Z_t$, $\hat{Z}_t^{tar} = Z_t^{tar} + \Delta Z_t$). If we simply remove it, it causes huge reconstruction errors. This makes the Latent Feature Interaction module fail completely. In contrast, by using "Stepwise Correction", we successfully isolate the effects of other modules.
>
> **New Ablation Study isolating the target trajectory:** Furthermore, to clarify the true effect of $\hat{Z}_t^{tar} = Z_t^{tar} + \Delta Z_t$, we provide an extra ablation study in the table below. Here, we only apply Direct Alignment to the source path ($\hat{Z}_t^{src} = Z_t^{src} + \Delta Z_t$) and leave the target trajectory unchanged ($\hat{Z}_t^{tar} = Z_t^{tar}$). We will include this new ablation in the final version.
>
> | **Component**            | **Structure ↓** | **PSNR ↑** | **LPIPS ↓** | **MSE ↓** | **SSIM ↑** | **CLIP （Whole） ↑** | **CLIP （Edited） ↑** |
> | ------------------------ | --------------- | ---------- | ----------- | --------- | ---------- | -------------------- | --------------------- |
> | **w/o mask,  tar_align** | 44.64           | 20.59      | 184.11      | 115.60    | 76.63      | 25.76                | 22.66                 |
> | **w/o mask**             | **21.93**       | **24.70**  | **102.92**  | **56.76** | **85.76**  | **25.89**            | **22.74**             |
>
> **Weakness 4: The trade-off evaluation and 2D scatter plots:**
>
> Thank you for this constructive suggestion! We have provided the requested 2D scatter plots for both the FLUX and SD3.5 backbones in the following [anonymous link](https://ibb.co/5XYg3fyd). As the figures clearly show, the results confirm that DirectEdit achieves the optimal balance. We will include these studies in the main paper.
>
> **Weakness 5: User Study and comparison with FlowAlign.**
>
> Thank you for these suggestions.
>
> **User Study:** We conducted a user study comparing the top-performing methods, with results shown in the following [anonymous link](https://ibb.co/VdspXGF). In this study, we designed evaluations covering various editing categories. For each category, we randomly sampled 20 test images from different methods. The volunteers were asked to choose their preferred results based strictly on two criteria: text alignment and background preservation. We collected over 80 valid responses in total. As the results show, DirectEdit received the highest user preference.
>
> **Comparison with FlowAlign:** We thank the reviewer for pointing out the recent FlowAlign method, which is a concurrent work at ICLR 2026. We provide a direct quantitative comparison on PIE-Bench in the following [anonymous link](https://ibb.co/vCJHZs8k). We will add these studies to the final version.

---

> > ### Author Rebuttal · Reviewer_RCwP · 2026-04-01
> >
> > Thanks for the detailed response. My concerns have been largely addressed.
> >
> > However, I still have a little question regarding the 'w.o. alignment' ablation, the study would be more convincing if you added a setting that leave both the source path ($\hat{Z}_t^{src} = Z_t^{src}$) and target path ($\hat{Z}_t^{tar} = Z_t^{tar}$) both unchanged" will make the ablation study more convincing.

---

> > > ### Author Response · Authors · 2026-04-01
> > >
> > > Dear Reviewer RCwP,
> > >
> > > We sincerely thank you for your continued engagement and this valuable suggestion. As requested, we provide the ablation results where both the source path ($\hat{Z}_t^{src} = Z_t^{src}$) and the target path ($\hat{Z}_t^{tar} = Z_t^{tar}$) remain unchanged in the table below:
> > >
> > > | Component                  | Structure ↓ | PSNR ↑    | LPIPS ↓   | MSE ↓     | SSIM ↑    | CLIP (Whole) ↑ | CLIP (Edited) ↑ |
> > > | :------------------------- | :---------- | :-------- | :-------- | :-------- | :-------- | :------------- | :-------------- |
> > > | Vanilla                    | 75.95       | 16.81     | 276.29    | 332.65    | 66.54     | 23.57          | 21.79           |
> > > | w/o alignment (both paths) | 74.95       | 16.66     | 276.50    | 338.42    | 66.40     | 23.49          | 21.35           |
> > > | **Ours**                   | **17.94**   | **32.63** | **35.45** | **25.05** | **93.49** | **25.39**      | **22.45**       |
> > >
> > > As shown, the performance drops significantly under this "both paths unchanged" setting. The results closely match the Vanilla baseline, which aligns with our expectations. As we analyzed in our response to Weakness 3, without a reliable inversion component, the severe accumulation of approximation errors simply causes all modules to fail. We will include these ablation settings in the final version to make our study more convincing. We hope this experiment fully addresses your concern, and we would be grateful if you would consider updating your score. Thanks again for your time and constructive suggestions!
> > >
> > > Best regards,
> > >
> > > Authors of paper #10252

---

### Official Review · Reviewer_JwcN · 2026-03-12

**Soundness:** 2
**Presentation:** 3
**Significance:** 2
**Originality:** 3
**Overall Recommendation:** 4
**Confidence:** 4

**Summary:**

The paper proposes DirectEdit, a training-free image editing framework for rectified-flow (RF) text-to-image transformers. This framework eliminates step-level reconstruction errors without requiring additional neural function evaluations. The core method is a direct alignment scheme. During forward sampling, the approach shifts the current latent by the per-step residual recorded during inversion and evaluates the velocity at the next-step inversion point. This process guarantees exact stepwise alignment of the reconstruction trajectory to the inversion path under Euler discretization. Building upon this alignment, the authors integrate an attention value-feature injection mechanism and a multi-branch mask-based latent blending strategy guided by MLLM and SAM. These components balance image fidelity and editability. The evaluation demonstrates strong performance on PIE-Bench using FLUX and SD3.5 backbones.

**Compliance With Llm Reviewing Policy:**

Affirmed.

**Final Justification:**

The authors provided clear explanations during the rebuttal process, which successfully resolved my initial concerns. Therefore, I recommend acceptance.

**Key Questions For Authors:**

1. How sensitive is performance to the timestep schedule and to using non-Euler solvers (e.g., Heun, RF‑Solver)? Does direct alignment extend naturally to these settings?

2. Can the proposed algorithm be applied to more precise image editing tasks? Please see weakness 1.

3. For fairness, did any baselines receive comparable region masks or blending mechanisms? If not, can you provide an ablation of DirectEdit “without MLLM/SAM” versus “with” on PIE-Bench?

**Limitations:**

Yes

**Strengths And Weaknesses:**

## Strength
1. The paper is well-structured. Figure 2 and Figure 3 provide intuitive visual explanations of the drift problem and the proposed solution. The algorithm is described clearly with equations and a pseudocode listing.
2. The core insight is sound. The direct alignment trick that adds the stored inversion residual ΔZt to the current latent before velocity evaluation shifts the evaluation point to Zt+1^inv and enforces step-level exactness under Euler. This simple yet powerful fix avoids extra NFEs.
3. Experiments on PIE-Bench with FLUX and SD3.5 as the backbone are detailed. The reconstruction analysis in Figure 5 supports the claim of near-zero step-level error.

## Weakness
1. Despite the algorithmic insights, focusing solely on the standard image editing task offers limited novelty compared to prior academic methods. Furthermore, the overall performance lags significantly behind contemporary large-scale models such as GPT and Gemini. While I do not expect the framework to match industry-level performance, I suggest the authors evaluate their proposed algorithm on more specialized tasks. For example, applying the method to image editing that requires large shape transformations [1][2] or editing guided by precise spatial control signals rather than text alone [3] would better demonstrate the utility of the approach.
2. The experimental setting may lack fairness. The reliance on an external MLLM (Qwen3-VL-Plus) and SAM for masks could give DirectEdit an advantage in background preservation compared to baselines that do not use such guidance. Fairness controls, such as evaluating with and without masks or providing baselines with comparable mask support, are limited.
3. The evaluation relies heavily on PIE-Bench metrics. More grounded assessments, such as the MCQ-based functional correctness and mask-aware preservation of GIE-Bench [4], and the lacking user studies, would strengthen the claims regarding edit correctness versus over-editing and under-editing.
4. Minor typographical/formatting issues (e.g., "Vallina," "Multi-branch") and occasional notation inconsistencies may distract. The mask-generation pipeline would benefit from clearer failure and latency statistics. Furthermore, architecture-specific injection rules, such as which blocks to inject, are presented as empirical choices. A brief rationale or a sensitivity analysis beyond t_inj would improve the manuscript.


[1] Long, Zeqian, et al. "Follow-your-shape: Shape-aware image editing via trajectory-guided region control." arXiv preprint arXiv:2508.08134

[2] Zhu, Tianrui, et al. "Kv-edit: Training-free image editing for precise background preservation." ICCV 2025

[3] Shin, Joonghyuk, et al. "Instantdrag: Improving interactivity in drag-based image editing." SIGGRAPH Asia 2024

[4] Qian, Yusu, et al. "Gie-bench: Towards grounded evaluation for text-guided image editing." arXiv preprint arXiv:2505.11493

---

> ### Author Rebuttal · Authors · 2026-03-31
>
> We sincerely appreciate your time and effort in reviewing our paper. We hope our following point-to-point responses can address your concerns.
>
> **Weakness 1: Comparison to large-scale models and evaluation on specialized tasks.**
>
> Thank you for the constructive comments. DirectEdit is a training-free, text-guided model, which operates under a completely different paradigm with industry-level models. Regarding the suggested baselines for specialized tasks: *Follow-your-shape* has not yet released its dataset, and *Kv-edit* evaluates on the exact same PIE-Bench dataset used in our main study. *Instantdrag* target point-based editing, which is not the core focus of our work. Due to the time limit, we would like to further explore the potential of DirectEdit on these tasks in future work.
>
> To further demonstrate our method's strength on complex editing tasks, we conducted an additional quantitative evaluation on a 100-image, high-resolution subset of *GIE-Bench*. This evaluation includes the metrics you suggested, with results shown in the following [anonymous link](https://ibb.co/R4V7WvTQ). Furthermore, we kindly ask the reviewer to refer to **Appendix E (Figure 10)** in our paper. It demonstrates our framework's ability to handle complex, non-rigid shape transformations on high-resolution real-world images.
>
> **Weakness 2: The experimental setting may lack fairness.**
>
> Thank you for raising this concern. To directly address the fairness concern and highlight the effectiveness of Direct Alignment, we kindly refer you to the **"w/o blending"** ablation study already provided in **Table 3** of our paper. Building on this, we provide a direct quantitative comparison in the following [anonymous link](https://ibb.co/vCJHZs8k). It compares DirectEdit *without* the MLLM/SAM masking mechanism against the baselines on PIE-Bench. We will include these comparisons in the final version.
>
> **Weakness 3: Evaluation on GIE-Bench and User Study.**
>
> Thank you for the suggestion. Please refer to our response to Weakness 1 for the evaluation on GIE-bench. We also conducted a user study comparing the top-performing methods, with results shown in the following [anonymous link](https://ibb.co/VdspXGF). In this study, we designed evaluations covering various editing categories. For each category, we randomly sampled 20 test images from different methods. The volunteers were asked to choose their preferred results based strictly on two criteria: text alignment and background preservation. We collected over 80 valid responses in total. As the results show, DirectEdit received the highest user preference.
>
> **Weakness 4: Typos/formatting issues, mask statistics, and injection rules**
>
> We sincerely thank you for carefully reading our paper and pointing out these details. We will thoroughly proofread the manuscript to correct all typos, formatting issues, and notation inconsistencies in the final version to improve readability.
>
> **Regarding mask generation failures and latency:** The mask generation pipeline is stable in most cases. However, for objects that are hard to segment accurately (like fluffy animals), it might sometimes cause unintended edge preservation. To fix this, as mentioned in **Appendix D** of our paper, we use a small dilation operation to smooth the mask boundaries. We demonstrate the specific effects at the following [anonymous link](https://ibb.co/pjhN5Brp). Regarding latency, the pipeline is very efficient. It takes only about 1.3 seconds per image on average and the mask can be reused across multiple editing turns.
>
> **Regarding the rationale for injection rules:** The decision of "which blocks to inject" has been well-studied and validated in previous works (e.g., *FTEdit*, *RFEdit*). Therefore, we followed their settings. This choice mainly affects the selection of the injection step, $t_{inj}$. We provide a sensitivity analysis for $t_{inj}$ in the following [anonymous link](https://ibb.co/994TmDK0). Additionally, please refer to **Figure 6** in our paper to understand how this parameter directly impacts the editing results.
>
> **Question 1: Sensitivity analysis for timestep and non-Euler solvers.**
>
> We provide a sensitivity analysis for the denoising timesteps in the following [anonymous link](https://ibb.co/mCF3N1GF). The results show that DirectEdit is highly robust across different timestep settings. Furthermore, Direct Alignment extends naturally and seamlessly to non-Euler solvers. The quantitative results are shown in the following [link](https://ibb.co/LT4myNf). We will include these studies in the revised manuscript.
>
> **Question 2: Application to more precise image editing tasks.**
>
> Yes, DirectEdit can be applied to more precise image editing tasks. Please refer to our response to Weakness 1.
>
> **Question 3: Baseline mask configurations and ablation without MLLM/SAM.**
>
> We thank the reviewer for this important question. We kindly refer the reviewer to our detailed response to Weakness 2.

---

> > ### Author Rebuttal · Reviewer_JwcN · 2026-04-01
> >
> > Thanks for the detailed response from the authors; my concerns are addressed, so I will keep my positive score.

---

> > > ### Author Response · Authors · 2026-04-01
> > >
> > > We are glad that the reviewer's concerns have been fully addressed. We sincerely thank the reviewer for their time and positive support for our work.
> > >
> > > Authors of paper #10252

---

### Decision · Program_Chairs · 2026-04-30

**Decision:**

Accept (regular)

**Comment:**

The submission received all positive ratings of 3 WA's and 1 A. Reviewers generally liked the overall approach and found the key insights useful, with good results and well written. Detailed concerns and questions were also raised, but they were also mostly clarified by the authors to the satisfaction of the reviewers. The AC agrees with the reviewers in recommending accept.